# Commensal lifestyle regulated by a negative feedback loop between *Arabidopsis* ROS and the bacterial T2SS

Frederickson Entila[1,2], Xiaowei Han [1,3,4], Akira Mine [5,6], Paul Schulze-Lefert [2] & Kenichi Tsuda [1,2,3,4] ✉

Despite the plant health-promoting effects of plant microbiota, these assemblages also comprise potentially detrimental microbes. How plant immunity controls its microbiota to promote plant health under these conditions remains largely unknown. We find that commensal bacteria isolated from healthy *Arabidopsis* plants trigger diverse patterns of reactive oxygen species (ROS) production dependent on the immune receptors and completely on the NADPH oxidase RBOHD that selectively inhibited specific commensals, notably *Xanthomonas* L148. Through random mutagenesis, we find that L148 *gspE*, encoding a type II secretion system (T2SS) component, is required for the damaging effects of *Xanthomonas* L148 on *rbohD* mutant plants. *In planta* bacterial transcriptomics reveals that RBOHD suppresses most T2SS gene expression including *gspE*. L148 colonization protected plants against a bacterial pathogen, when *gspE* was inhibited by ROS or mutation. Thus, a negative feedback loop between *Arabidopsis* ROS and the bacterial T2SS tames a potentially detrimental leaf commensal and turns it into a microbe beneficial to the host.

In nature, plants host diverse microbes called the plant microbiota[1]. While the plant microbiota collectively contributes to plant health, they comprise microorganisms ranging from mutualistic to commensal, and pathogenic microbes[2]. The property of microbes as mutualistic, commensal, and pathogenic depends on the host and environmental condition[3,4]. Thus, the plant microbiota is not simply a collection of beneficial microbes, but various factors affect the property of microbes within the plant microbiota, which consequently determines plant health.

Upon recognition of microbial molecules, plants activate a battery of immune responses[5]. In the first layer of immunity, known as pattern-triggered immunity (PTI), plasma membrane-localized pattern recognition receptors (PRRs) recognize microbe-associated molecular patterns (MAMPs). For instance, the PRR FLAGELLIN SENSING 2 (FLS2) and EF-TU RECEPTOR (EFR) sense the bacteria-derived oligopeptides flg22 and elf18, respectively, in *Arabidopsis thaliana*[6,7]. BRI1-ASSOCIATED RECEPTOR KINASE 1 (BAK1) and its close homolog BAK1-LIKE 1 (BKK1) function as co-receptors for LRR-RLK-type PRRs such as FLS2 and EFR[8]. The LysM-RLK CHITIN ELICITOR RECEPTOR KINASE 1 (CERK1) is an essential co-receptor for fungal chitin and bacterial peptidoglycans[9]. Activated PRRs trigger various immune responses such as the production of reactive oxygen species (ROS),

[1]National Key Laboratory of Agricultural Microbiology, Hubei Hongshan Laboratory, Hubei Key Laboratory of Plant Pathology, College of Plant Science and Technology, Huazhong Agricultural University, Wuhan 430070, China. [2]Department of Plant Microbe Interactions, Max Planck Institute for Plant Breeding Research, Carl-von-Linne-Weg 10, Cologne 50829, Germany. [3]Shenzhen Institute of Nutrition and Health, Huazhong Agricultural University, Wuhan 430070, China. [4]Shenzhen Branch, Guangdong Laboratory of Lingnan Modern Agriculture, Genome Analysis Laboratory of the Ministry of Agriculture and Rural Affairs, Agricultural Genomics Institute at Shenzhen, Chinese Academy of Agricultural Sciences, Shenzhen, Guangdong 518120, China. [5]JST PRESTO, Kawaguchi-shi, Saitama 332-0012, Japan. [6]Laboratory of Plant Pathology, Graduate School of Agriculture, Kyoto University, Kyoto 606-8502, Japan. ✉e-mail: tsuda@mail.hzau.edu.cn

calcium influx, MAP kinase activation, transcriptional reprogramming, and the production of defense phytohormones and specialized metabolites[10]. PTI contributes not only to pathogen resistance but also to the maintenance of healthy microbiota as evidenced by dysbiosis and disease symptoms observed on leaves of *A. thaliana* genotypes with severely impaired PTI responses[11,12]. However, the molecular mechanism by which PTI-associated immune responses regulate microbial pathogens and maintain healthy microbiota remains unclear.

One prominent PTI output involves activation of the plasma membrane-localized NADPH oxidase RESPIRATORY BURST OXIDASE HOMOLOG D (RBOHD), which produces the ROS $O_2^-$ in the extracellular space, which can then be readily converted to $H_2O_2$ via superoxide dismutase in the apoplast[13]. Extracellular ROS can be sensed by a plasma membrane-localized sensor and can be translocated into the cell to mediate plant immune responses[14]. Extracellular ROS can also directly exert cellular toxicity on microbes[15]. ROS functions in regulating not only resistance against pathogens, but also the composition and functions of the plant microbiota. For instance, RBOHD-mediated ROS production inhibits Pseudomonads in the *A. thaliana* rhizosphere[16]. *RBOHD* also prohibits dysbiosis in *A. thaliana* leaves by suppressing *Xanthomonas* L131 and the close strain L148, as indicated by their profuse colonization associated with disease onset on *rbohD* mutant plants upon inoculation with synthetic bacterial communities[17]. Plant RBOHD-mediated ROS induces the production of the phytohormone auxin in the beneficial bacterium *Bacillus velezensis* and promotes root colonization in *A. thaliana*[18]. These studies exemplify the importance of RBOHD-mediated ROS production in the regulation of plant microbiota. However, how ROS specifically regulates microbial metabolism and growth remains unknown. Furthermore, while ROS exhibits general cell toxicity to organisms, not all microbes are sensitive to plant-produced ROS. For instance, the growth of the bacterial pathogen *Pseudomonas syringae* pv. *tomato* DC3000 (*Pto*) was not affected by mutation in *RBOHD* in *A. thaliana*[19]. This indicates that ROS exerts differential actions on microbes, but the basis for this selectivity needs to be explored.

Secretion systems are crucial for bacterial pathogens to efficiently infect the host plant through the secretion of effector proteins, among which the type III secretion system (T3SS) has been well documented as the essential pathogenicity component of many phytopathogenic bacteria[20]. The key function of T3SS is to introduce type III effectors (T3Es) directly into the host cell, thereby suppressing plant immunity and promoting virulence[20]. Some nitrogen-fixing rhizobacteria also utilize T3SS to promote symbiosis with their legume host[21,22]. A number of T3Es have been identified to be recognized by the intracellular nucleotide-binding domain leucine-rich repeat receptors (NLRs), activating effector-triggered immunity[23]. These indicate the paramount significance of T3SS for the interaction between host and bacteria. In addition to T3SS, the type II secretion system (T2SS) has been shown to be necessary for the pathogenesis of many phytopathogenic bacteria and functions to secrete enzymes to degrade host barriers and promote virulence[20]. Interestingly, the root commensal *Dyella japonica* MF79 requires the T2SS components *gspD* and *gspE* to release immune-suppressive factors that help the root colonization of a non-immune suppressive commensal in *A. thaliana*[24]. However, whether and how plant immunity controls T2SS activity of its microbiota remains unknown.

In this study, we investigated the impact of *A. thaliana* immune responses on commensal bacteria isolated from healthy *A. thaliana* plants with a focus on RBOHD-mediated ROS. Using a bacterial random mutagenesis screen and *in planta* bacterial transcriptomics, we revealed that RBOHD-mediated ROS directly suppresses T2SS of a potentially harmful *Xanthomonas* L148, which makes *Xanthomonas* L148 a commensal. Moreover, this "tamed" *Xanthomonas* increased host resistance against the bacterial pathogen *Pto*.

## Results

### Different commensal bacteria trigger diverse ROS production patterns via distinct mechanisms

We investigated variations in immune responses triggered by the colonization of different commensal bacteria in *A. thaliana* leaves with ROS production as the readout. First, we measured ROS production in leaves of *fls2*, *efr*, *cerk1*, *fls2 efr cerk1* (*fec*), *bak1 bkk1 cerk1* (*bbc*), and *rbohD* mutants as well as Col-0 wild-type plants in response to the MAMPs flg22, elf18, and chitin heptamer. ROS production was dependent on the corresponding (co)receptor and *RBOHD*, indicating the suitability of our experimental system (Fig. 1a and Supplementary Fig. S1). Next, we measured ROS production in leaves of the same mutant panel in response to taxonomically diverse 20 live and heat-killed commensal bacterial strains (Fig. 1d) that were previously isolated from healthy *A. thaliana* leaves and roots as well as soil[25] and that were used for plant-bacterial co-transcriptomics[26]. These commensal bacteria triggered diverse ROS production patterns in a *RBOHD*-dependent manner which is consistent with the recent finding using heat-killed cells of leaf-isolated strains[17]. For instance, both live and heat-killed *Pseudomonas* L127 triggered ROS production with the heat-killed bacteria eliciting stronger ROS production, which is a general trend for all commensal bacterial strains (Fig. 1c and Supplementary Fig. S2). On the other hand, only heat-killed but not live *Burkholderia* L177 triggered ROS production. Further, neither the live nor the heat-killed *Flavobacterium* R935 triggered ROS production (Fig. 1d and Supplementary Fig. S2). We observed neither obvious phylogenetic signatures predictive for the capability to induce ROS, nor of the tissue of origin from which these commensals were isolated. We also observed different dependencies of commensal bacteria-induced ROS on the MAMP (co)receptors. For instance, ROS production by both live and heat-killed *Exiguobacterium* L187 was dependent on *EFR* but not *FLS2* and *CERK1*. This *EFR* dependency for commensal bacteria-induced ROS production was observed for other strains, since many of which had reduced or lost ROS burst for *efr* leaf disks but considerable ROS signals were detected for *fls2* and *cerk1* (Supplementary Fig. S2). These results suggest that the recognition of EF-Tu-derived peptides via EFR is the primary mechanism for ROS production by commensal bacteria in *A. thaliana* leaves. However, there were commensal bacteria such as *Pseudomonas* L127 that stimulated ROS in *fec* and *bbc* mutant plants (Supplementary Fig. S2), indicating that MAMPs other than flg22, elf18, and peptidoglycans are responsible for ROS production induced by commensal bacteria in some cases.

### Plant-derived ROS differentially affects the colonization of commensal bacteria

We found that ROS production by all live and heat-killed commensal bacteria was completely dependent on *RBOHD*, indicating that RBOHD is mainly responsible for plant ROS production triggered by these commensal bacteria. Plant-produced ROS via RBOHD can affect the colonization of commensal bacteria. We then determined total and endophytic bacterial titers of different commensals in leaves of Col-0 wild-type and *rbohD* as well as *fls2*, *efr*, *fec*, and *bbc* mutant plants. We grew plants on agar plates for 14 days and flood-inoculated with individual commensal bacteria followed by the determination of bacterial titer (Fig. 1b, Supplementary Fig. S3). To our surprise, while we did observe increased colonization of some commensal bacteria in some of the MAMP (co)receptor mutants compared with Col-0 wild-type plants, we were largely unable to detect any impact of the *rbohD* mutation on either total or endophytic commensal colonization (Fig. 1d and Supplementary Fig. S3). Also, there is no significant relationship between the ROS immunogenicity and the colonization capacity of the commensal bacteria (Supplementary Fig. S4). By contrast, both total and endophytic colonization of *Xanthomonas* L148 was dramatically increased in *rbohD* mutant compared with Col-0 wild-type plants (Fig. 1d and Supplementary Fig. S3), suggesting that

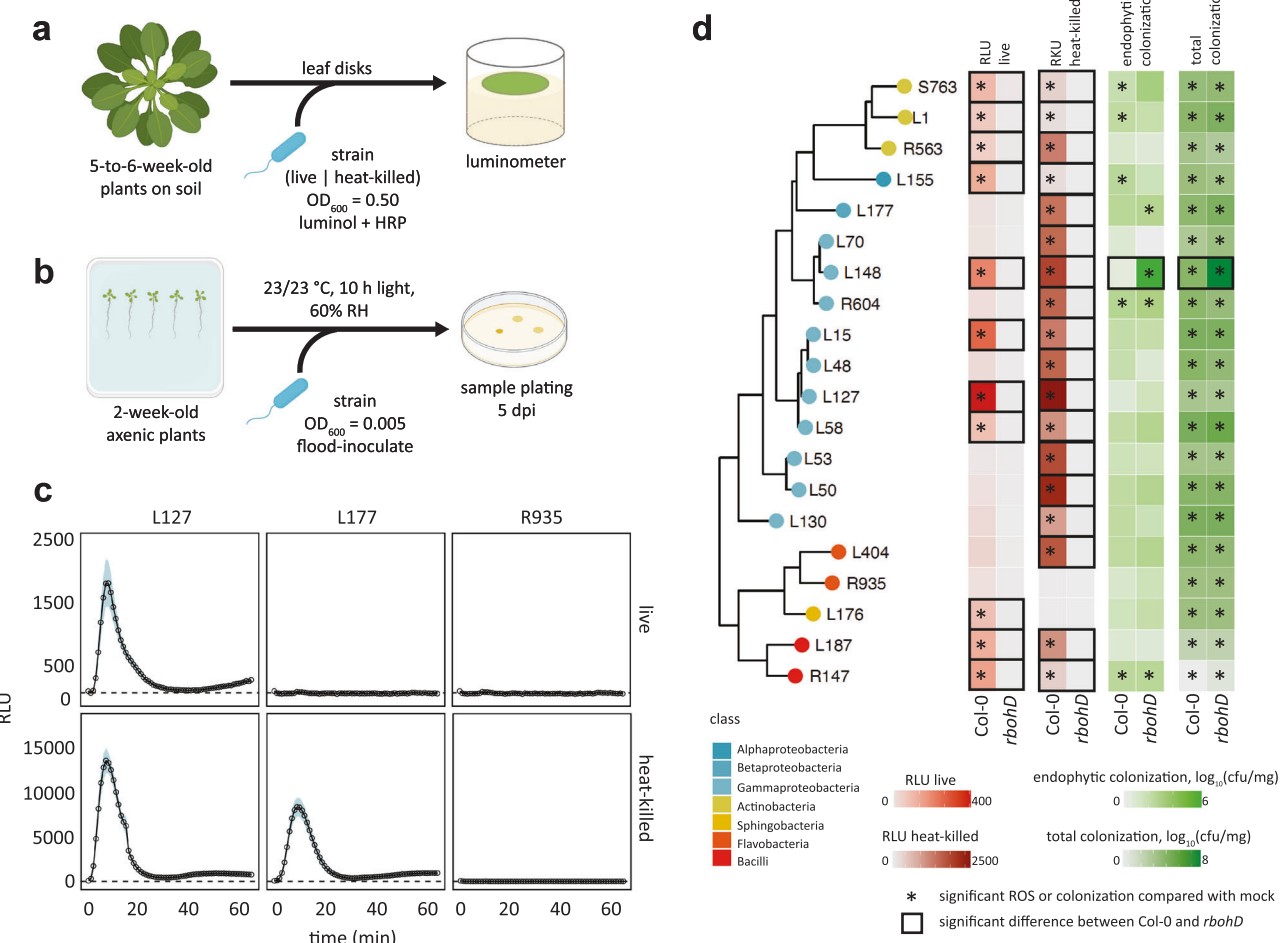

**Fig. 1 | Immunogenic and colonization profile of microbiota members in mono-associations.** Schematic diagram of ROS burst assay (**a**) in leaf discs from 5 to 6-week-old Col-0 plants treated with live or heat-killed bacterial cells (OD$_{600}$ = 0.5) and colonization capacities (**b**) of the microbiota members upon flood inoculation (OD$_{600}$ = 0.005) of 2-week-old Col-0 plants at 5 dpi. **c** ROS burst profile of representative strains with varying behaviors: immune-active, -evasive, and -quiescent, for *Pseudomonas* L127, *Burkholderia* L177, and *Flavobacterium* R935, respectively (see Supplementary Fig. S2 for the full ROS burst profiles) wherein results are shown as line graphs using Locally Estimated Scatter Plot Smoothing (LOESS) with error bars and shadows indicating the standard errors of the mean. **d** Phylogenetic relationship of the selected microbiota members and the heatmap representation of their corresponding ROS burst profiles using live and heat-killed cells, and their respective colonization capacities in leaves of Col-0 and *rbohD* plants; * indicates significant within-genotype difference of the trait between mock and the bacterial strain in question; □ indicates significant within-strain difference of the trait between Col-0 and *rbohD* plants (two-sided ANOVA with *post hoc* Tukey's test, *P* ≤ 0.05). Experiments were repeated at least two times each with 8 biological replicates for ROS assay and 3–4 biological replicates for colonization assays (See Supplementary Fig. S3 for the full colonization profiles and Supplementary Data 1 for detailed descriptions of the strains included). Some illustrations were created with BioRender.

RBOHD-mediated ROS suppresses *Xanthomonas* L148 colonization, consistent with a recent finding[17].

### *Xanthomonas* L148 is detrimental to *rbohD* mutant but not Col-0 wild-type plants

Leaf colonization of *rbohD* mutant plants with live *Xanthomonas* L148 led to host mortality within 5 days post inoculation (dpi), in contrast to asymptomatic wild-type Col-0 plants (Fig. 2a). In an orthogonal system, we infiltrated leaves with *Xanthomonas* L148 and observed disease-like symptoms only in *rbohD* after 3 dpi (Fig. 2b–d). As *Xanthomonas* L148 activated ROS burst in Col-0 leaves, but not in *rbohD*, *Xanthomonas* L148 pathogenicity might be suppressed by the ROS pathway (Fig. 2e). Furthermore, *Xanthomonas* L148 not only persisted on the leaf surface but aggressively colonized the apoplast of *rbohD* mutants compared with Col-0 at 3 dpi (Fig. 2f). Together, *Xanthomonas* L148 is potentially pathogenic and its deleterious effect depends on the absence of *RBOHD*.

### *Xanthomonas* L148 in vitro growth is largely unaffected by ROS

Due to their highly reactive nature, ROS can oxidize bacterial components, which can lead to extensive cellular damage. This might explain why *Xanthomonas* L148 is pathogenic to *rbohD* mutant but not to Col-0 wild-type plants. We tested the sensitivity of *Xanthomonas* L148 to ROS compounds by instantaneous in vitro exposure to H$_2$O$_2$ or O$_2^{-1}$. To our surprise, *Xanthomonas* L148 seemed to tolerate acute treatments with ROS and retained viability up to H$_2$O$_2$ and O$_2^{-1}$ concentrations of 1 mM (Supplementary Fig. S5a, b). Similar findings were obtained when a ROS-generating compound, paraquat (PQ, Supplementary Fig. S5c), was used. It can be argued that the adverse effects of ROS in vitro can only be observed upon continuous ROS treatment. However, we did not observe any significant effects on the growth rates of *Xanthomonas* L148 upon chronic exposure to PQ (Supplementary Fig. S5d). This suggests that the rampant proliferation of *Xanthomonas* L148 in *rbohD* plants is not due to the direct microbiocidal effects of ROS but other mechanisms.

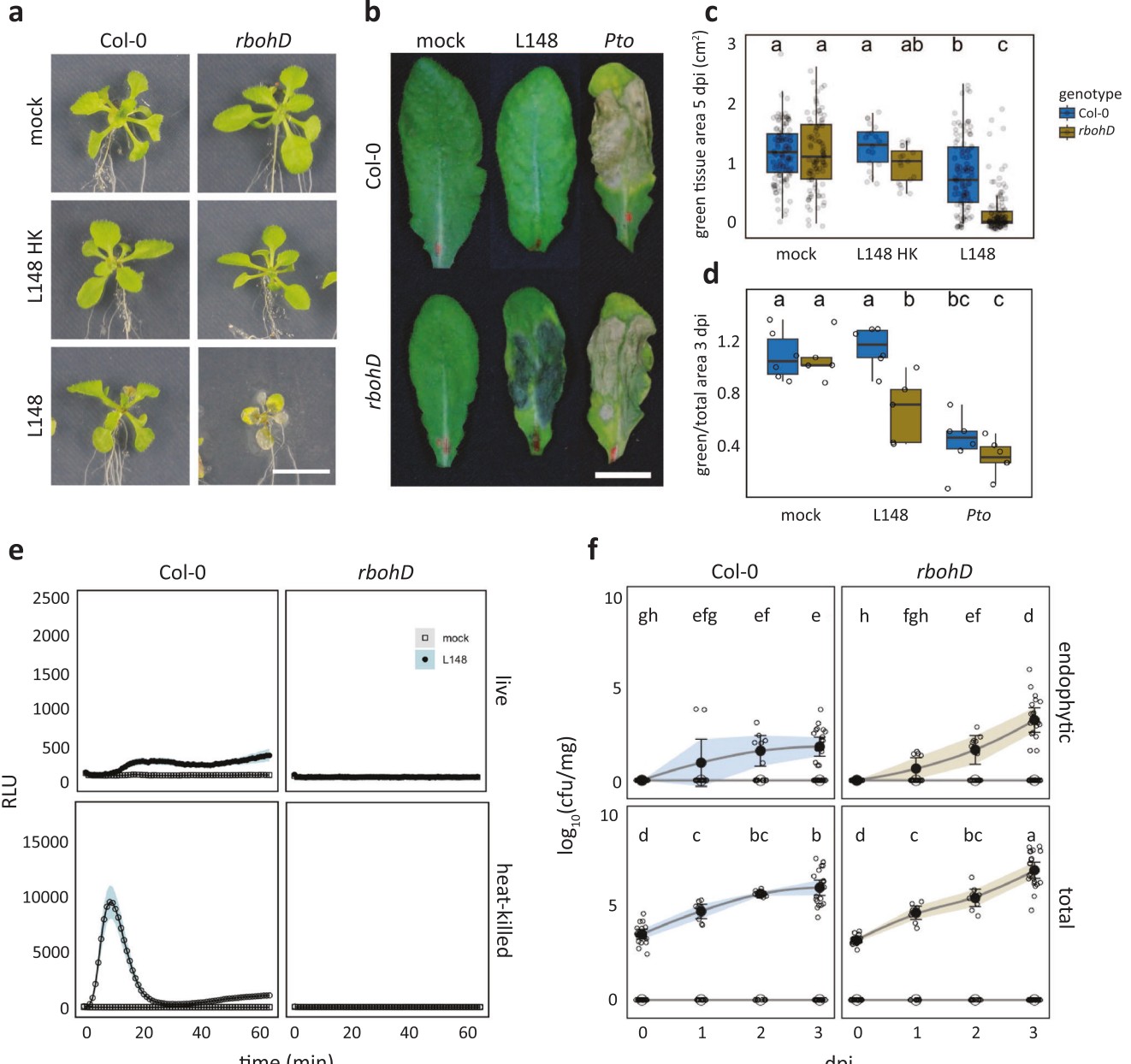

**Fig. 2 | *Xanthomonas* L148 is detrimental to *rbohD* mutant but not to Col-0 wild-type plants. a–c** Representative images (**a**) and quantification of green tissue area (**c**) as the plant health parameter. 14-day-old Col-0 and *rbohD* plants grown on agar plates were flood-inoculated with mock and live and heat-killed (HK) *Xanthomonas* L148 (OD$_{600}$ = 0.005). Samples were taken at 5 dpi (4 independent experiments each with at least 5 biological replicates, $n$ = 114, 112, 19, 18, 115, and 114, from left to right). **b–d** Representative images (**b**) and quantification of percentage green tissue of leaves (**d**) hand-infiltrated with mock, *Xanthomonas* L148 and *Pto* (OD$_{600}$ = 0.2). Samples were taken at 3 dpi (2 independent experiments each with 3–4 biological replicates, $n \geq 6$). **e** ROS burst profile of leaf discs of 5–6-week-old Col-0 and *rbohD* plants treated with live and heat-killed *Xanthomonas* L148 (OD$_{600}$ = 0.5) (at least 4 independent experiments each with 8 biological replicates,

$n \geq 40$ and 32 for Col-0 and *rbohD* plants, respectively). **f,** Infection dynamics of *Xanthomonas* L148 upon flood inoculation of 14-day-old Col-0 and *rbohD* plants grown in agar plates (OD$_{600}$ = 0.005). Leaf samples were harvested at 0 to 3 dpi for total and endophytic compartments (2 independent experiments each with 3–4 biological replicates, $n$ = 14, 9, 9, and 21, for 0, 1, 2, and 3 dpi). Results in **c** and **d** are depicted as box plots with the boxes spanning the interquartile range (IQR, 25th to 75th percentiles), the mid-line indicates the median, and the whiskers cover the minimum and maximum values not extending beyond 1.5x of the IQR. Results in (**e**, **f**) are shown as line graphs using Locally Estimated Scatter Plot Smoothing (LOESS) with error bars and shadows indicating the standard errors of the mean. **c**, **d–f** two-sided ANOVA with *post hoc* Tukey's test. Different letters indicate statistically significant differences ($P \leq 0.05$).

## *Xanthomonas* L148 pathogenic potential is partially suppressed by the presence of other leaf microbiota members

*Xanthomonas* L148 was isolated from macroscopically healthy *A. thaliana* plants grown in their natural habitat, indicating that it is a constituent of the native leaf microbiota of *A. thaliana*. While *Xanthomonas* L148 was detrimental to *rbohD* mutant plants in a mono-association condition, it can be postulated that in a microbial

community setting, *Xanthomonas* L148 is disarmed and *rbohD* plants become asymptomatic. To test this, we constructed a synthetic bacterial community which consists of nine leaf-derived isolates that were found to be robust leaf colonizers and cover the major phyla of the native bacterial microbiota of leaves[27–29], which we refer to as LeafSC (Supplementary Fig. S6a, please see Supplementary Data 1 for the strain details). We also assessed the dose-dependency of the disease

onset by using different proportions of *Xanthomonas* L148 in relation to the entire LeafSC, with L148$_{P1}$ as a dose equivalent to that of each synthetic community member (*Xanthomonas* L148/LeafSC, 1:9), while L148$_{P9}$ is a dosage that is equal to the entire bacterial load of the synthetic community (*Xanthomonas* L148/LeafSC, 9:9). Flood inoculation of Col-0 and *rbohD* plants with the LeafSC did not result in any observable disease symptoms (Supplementary Fig. S6b and c). As expected, inoculation with *Xanthomonas* L148 resulted in substantial mortality of *rbohD* plants compared with Col-0 wild-type plants. The killing activity of *Xanthomonas* L148 was somewhat reduced in *rbohD* plants when other microbiota strains were present, but this counter effect was overcome when a higher dose of *Xanthomonas* L148 was used (Supplementary Fig. S6b and c). These findings imply that a functional leaf microbiota contributes to the partial mitigation of disease symptoms caused by *Xanthomonas* L148 in *rbohD* plants, possibly through niche occupancy, resource competition, or antibiosis.

### *Xanthomonas* L148::Tn5 mutant screening unveils genetic determinants of its pathogenic potential

*Xanthomonas* L148 is a conditional pathogen and its virulence is unlocked in the absence of *RBOHD* in the plant host. We aimed to identify the bacterial genetic determinants of this trait through a genome-wide mutant screening. We developed and optimized a robust high-throughput screening protocol (Fig. 3a, Supplementary Fig. S7a) and generated and validated a *Xanthomonas* L148 Tn5 mutant library (Supplementary Fig. S7b–d). Using the high-throughput protocol, this Tn5 mutant library was phenotyped for the loss-of-*rbohD* mortality. From 6,862 transposon insertional mutants, 214 candidate strains consistently failed to exert pathogenicity on *rbohD* mutant plants (Fig. 3b, See Supplementary Data 2 for the complete list of the candidate mutant strains). Most of the 214 strains did not exhibit significant defects in their in vitro growth parameters (growth rate, biofilm formation, and motility) in rich TSB medium or minimal XVM2 medium (Fig. 3c). We found that out of the 214 strains, only 124 had transposon insertions in genes with functional annotations. These strains were retested in a square plate agar format, and 18 bacterial mutants exhibited consistent loss-of-*rbohD* mortality (Fig. 3b). Out of these 18 strains, three showed very strong phenotypes, namely *gspE*::Tn5, *alaA*::Tn5, and *rpfF*::Tn5 (Fig. 3d–f). The candidate gene *gspE* encodes a core ATPase component of the T2SS; *alaA* encodes an alanine-synthesizing transaminase involved in amino acid metabolism; and *rpfF* encodes a synthase for diffusible signaling factor (DSF), a constituent of the quorum sensing machinery in bacteria (Fig. 3e).

### T2SS, amino acid metabolism, and quorum sensing underpin the conditional pathogenicity of *Xanthomonas* L148

We re-evaluated the candidate mutant strains using leaf-infiltration assays. The results showed that the disease progression required live *Xanthomonas* L148 as heat-killed bacteria did not elicit the same response (Fig. 4a). Consistent with the previous systems (high-throughput and square plate set-ups), the mutant strains lost their capacity to cause disease symptoms on *rbohD* mutant plants (Fig. 4). As shown before, wild-type *Xanthomonas* L148 exhibited increased colonization in both total and endophytic compartments of *rbohD* leaves. By contrast, *gspE*::Tn5 mutant exhibited comparable endophytic colonization and slightly lower total leaf colonization capacities to *Xanthomonas* L148 wild-type in Col-0 leaves, but failed to show increased colonization in *rbohD* compared with Col-0 plants in contrast to wild-type L148 (Fig. 4b). On the other hand, *alaA*::Tn5 mutants had a compromised colonization capacity in Col-0 plants, while *rpfF*::Tn5 mutant strains colonized *rbohD* leaves to a similar extent to wild-type *Xanthomonas* L148. Nonetheless, all of the mutant strains not only persisted but were able to actively colonize the leaf endosphere (Fig. 4b). This indicates that *gspE*::Tn5 mutant retains its endophytic

colonization ability, while its capacity to more efficiently colonize *rbohD* plants is compromised compared to wild-type L148. Correlation analysis revealed a negative relationship between host colonization and plant health, indicating that the observed leaf symptoms can be explained by the aggressive colonization of the wild-type strain (Fig. 4d).

None of the three mutant strains were defective in growth, biofilm production, or motility in rich TSB medium (Fig. 5a–c). Also, the mutant strains remained insensitive to PQ treatment, indicating retained tolerance to chronic ROS exposure (Fig. 5a). In vitro growth phenotypes were also unchanged in minimal XVM2 medium apart from an increase in biofilm production for *gspE*::Tn5 and *alaA*::Tn5 mutant strains (Fig. 5d). Secretion of extracellular enzymes acting on plant cell walls is a canonical strategy used by plant pathogens to breach the host's physical barriers[20]. Bacterial pathogens often utilize T2SS to deliver these enzymes into the apoplast of their plant host[30]. We conducted enzyme secretion plate assays to test the proficiency of these strains to degrade different substrates (carbohydrates, protein, and lipids). Wild-type *Xanthomonas* L148 was able to secrete extracellular enzymes that can degrade the proteinaceous compounds gelatin and non-fat dry milk and the carbohydrates pectin and carboxymethyl-cellulose. Notably, *gspE*::Tn5 mutant could not degrade or to a lesser extent, these substrates in contrast to the wild-type and the other mutant strains, indicating impaired secretion activities (Fig. 5e, f). This suggests that the lack of disease progression in *rbohD* plants with the *gspE*::Tn5 mutant strain can be explained by its inability to secrete extracellular enzymes to degrade the host plant cell walls via T2SS.

To gain insight into the evolution of the pathogenicity of *Xanthomonas* L148, available genomes of other Xanthomonadales members, including the potentially pathogenic close-relative *Xanthomonas* L131[17] and *Xanthomonas* L70 in the AtSPHERE[25], together with several *Xanthomonas* phytopathogens were interrogated for the occurrence of secretion systems and their potential CAZyme catalogs. In general, all Xanthomonadales strains encode both T1SS and T2SS genes (Supplementary Fig. S8a). The pathogenic and potentially pathogenic Xanthomonadales strains have expanded their CAZyme repertoire with proclivities for plant cell wall components: α-, β-glucans, α-mannans, arabinan, cellulose, hemi-cellulose, xylan, arabinoxylan, and pectin (Supplementary Fig. S8b, c). This indicates that though secretion systems are prevalent among the Xanthomonadales members, CAZyme repertoire expansion might be key feature of pathogenic and potentially pathogenic strains.

Because of the *in planta, ex planta*, and in vitro phenotypes, we focused on *gspE*::Tn5 mutant and characterized it extensively. To establish that *gspE* determines *rbohD*-dependent pathogenicity, we generated two independent *gspE* deletion mutant strains (Δ*gspE*_1 and Δ*gspE*_2) via homologous recombination. Both of the *gspE* deletion and the *gspE*::Tn5 mutants showed reduced secretion activities and failed to cause disease in *rbohD* plants (Fig. 6a, b). Taken together, *gspE*, an integral component of T2SS, is essential for *Xanthomonas* L148 pathogenicity on *rbohD* mutant plants.

### Plant ROS suppresses T2SS genes including *gspE* of *Xanthomonas* L148

*Xanthomonas* L148 pathogenicity is exerted in the absence of ROS through *RBOHD*, while our in vitro results do not indicate general cellular toxicity of ROS. Thus, it can be assumed that *RBOHD*-mediated ROS production suppresses the virulence of *Xanthomonas* L148. To gain insight into this, we conducted *in planta Xanthomonas* L148 bacterial transcriptome profiling for Col-0 and *rbohD* plants[26]. Plants were flood-inoculated with *Xanthomonas* L148 and shoots were sampled at 2 dpi, a time point at which bacterial titers were still indistinguishable; these later became significantly different between Col-0 and *rbohD* leaves at 3 dpi (Fig. 2f). Thus, with the bacterial

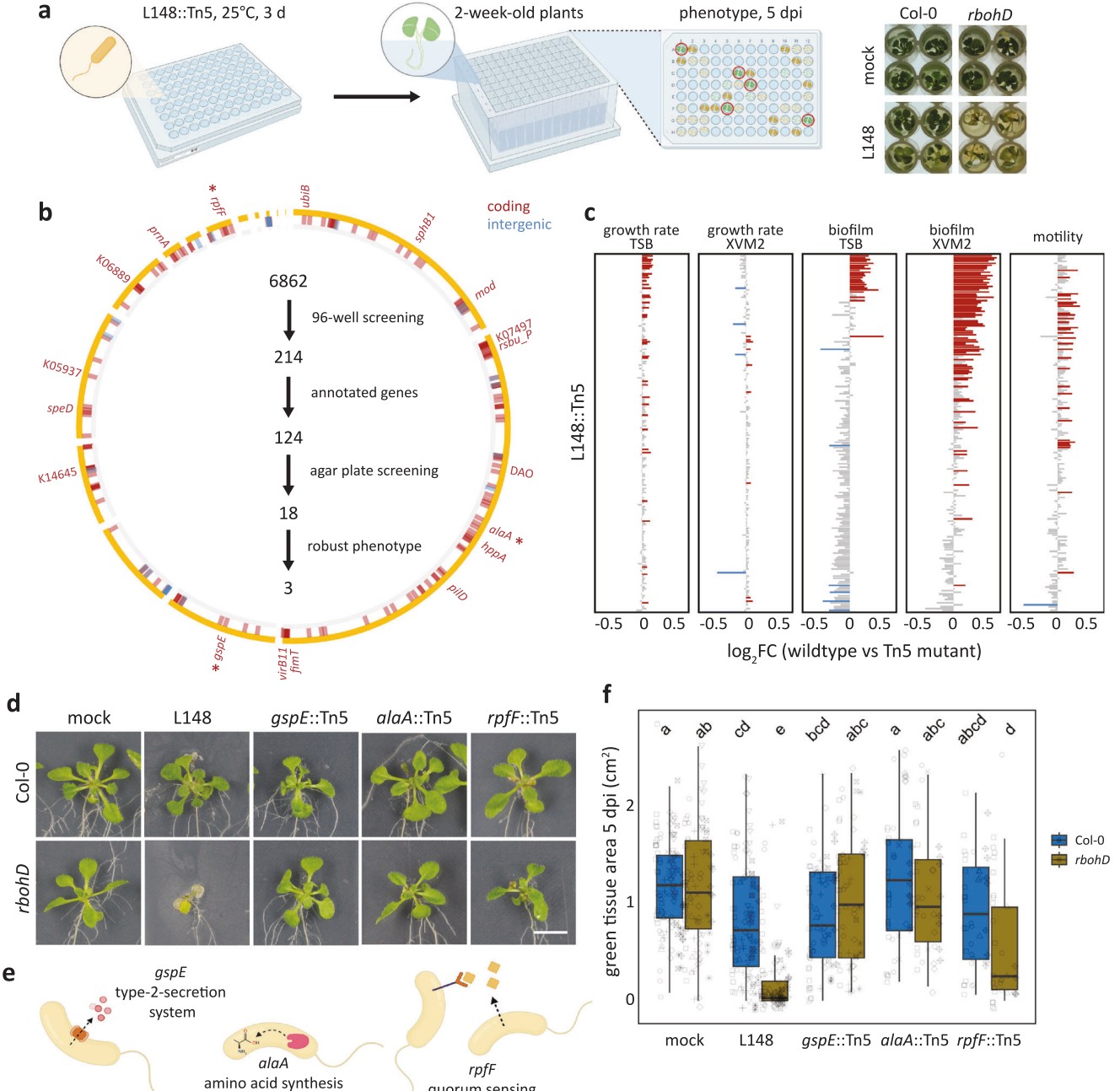

**Fig. 3 | *Xanthomonas* L148::Tn5 mutant screening unveils genetic determinants of its pathogenic potential. a** Schematic diagram of the optimized high-throughput genetic screening for the *Xanthomonas* L148::Tn5 mutant library. Bacterial strains were inoculated onto 2-week-old *rbohD* plants followed by phenotyping at 5 dpi. **b** Genomic coordinates of genes disrupted in the 214 *Xanthomonas* L148::Tn5 candidate strains. A total of 6862 *Xanthomonas* L148::Tn5 strains were screened for loss of *rbohD* killing activity in a 96-well high-throughput format (2 independent experiments). We identified 124 strains with functional annotations, which were subsequently screened using the agar plate format, resulting in 18 strains with robust phenotypes which are labeled. Finally, 3 strains, indicated with (*) were selected as the best-performing candidate strains. **c** In vitro phenotypes of the 214 candidate strains: growth rates, biofilm production, and motility in rich TSB medium; growth rates and biofilm production in a minimal XVM2 medium. Data from 2 independent experiments each with 2–3 biological replicates were

used for two-sided ANOVA with a *post hoc* Least Significant Difference (LSD) test. Red and blue bars indicate significantly higher or lower than the wild-type *Xanthomonas* L148 ($P \leq 0.05$), respectively. **d–f** Representative images (**d**) and quantification of green tissue area (**f**) as plant health parameter of Col-0 and *rbohD* plants flood mono-inoculated with *Xanthomonas* L148::Tn5 strains ($OD_{600} = 0.005$). Samples were harvested at 5 dpi. Data from at least 4 independent experiments each with 3–4 biological replicates ($n = 114, 112, 115, 114, 75, 64, 55, 53, 44, 40$, from left to right) were used for two-sided ANOVA with a *post hoc* Tukey's test. Different letters indicate statistically significant differences ($P \leq 0.05$). **e** Graphical representation of the functions of the candidate genes. Results in (**f**) are depicted as box plots with the boxes spanning the interquartile range (IQR, 25th to 75th percentiles), the mid-line indicates the median, and the whiskers cover the minimum and maximum values not extending beyond 1.5x of the IQR. Some of the illustrations were created using BioRender.

transcriptomes observed at this time point, one can exclude the possibility that the differences in expression are due to the different bacterial population densities known to affect bacterial transcriptome[31].

Principal component (PC) analysis revealed that *in planta Xanthomonas* L148 transcriptomes were distinct in Col-0 and *rbohD* plants (Fig. 7b). Statistical analysis revealed 2946 differentially expressed genes (DEGs) upon comparing *in planta* bacterial transcriptomes in

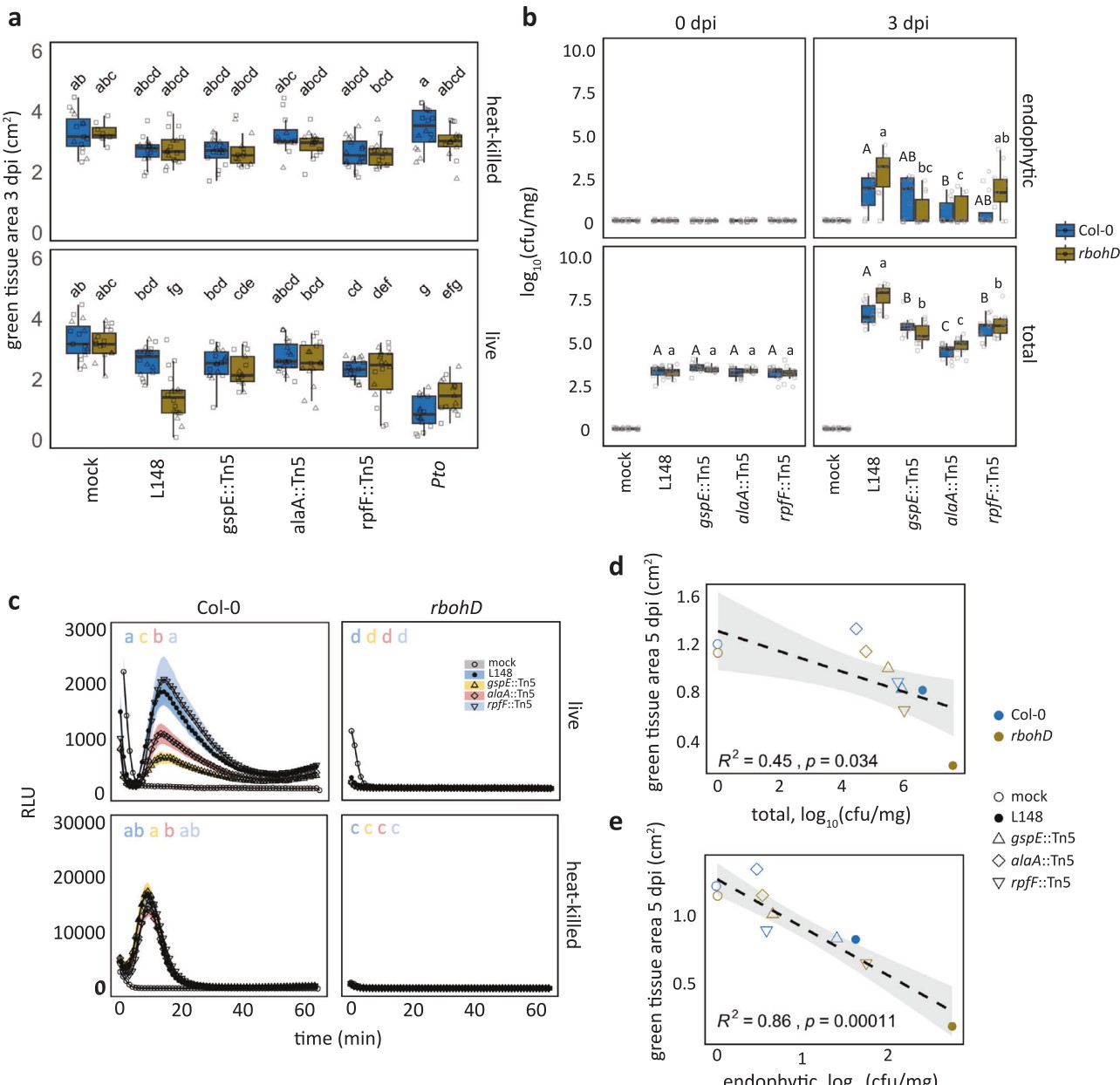

**Fig. 4 | T2SS, amino acid metabolism, and quorum sensing underpin conditional pathogenicity of *Xanthomonas* L148 in *rbohD* plants. a** Quantification of green tissue area of hand-infiltrated 5–6-week-old Col-0 and *rbohD* leaves with *Xanthomonas* L148::Tn5 mutant as well as wild-type strains using live and heat-killed cells as inoculum (OD$_{600}$ = 0.2). Samples were collected at 3 dpi (2 independent experiments each with 3–4 biological replicates, for heat-killed, *n* = 8, 15, 16, 16, 17, 14, 12, 15, 16, and 14 from left to right; for live cells, *n* = 15, 15, 16, 18, 18, 13, 19, 16, 18, and 18 from left to right). **b** Infection dynamics in axenic Col-0 and *rbohD* plants flood-inoculated with *Xanthomonas* L148::Tn5 mutant as well as wild-type strains (OD$_{600}$ = 0.005). Samples were harvested at 0 to 3 dpi for total and endophytic leaf compartments (2 independent experiments each with 3–4 biological replicates, *n* ≥ 24). **a** two-sided ANOVA with *post hoc* Tukey's test. Different letters indicate statistically significant differences (*P* ≤ 0.05). **b** two-sided ANOVA with post hoc Tukey's test implemented comparing bacterial strains within the genotype,

compartment, and dpi. Different letters (capital letters for Col-0, and small letters for *rbohD*) indicate statistically significant differences within the genotype, compartment, and dpi (*P* ≤ 0.05). Results in (**a**, **b**) are depicted as box plots with the boxes spanning the interquartile range (IQR, 25th to 75th percentiles), the mid-line indicates the median, and the whiskers cover the minimum and maximum values not extending beyond 1.5x of the IQR. **c**, ROS burst profile of leaf discs of 5–6-week-old Col-0 and *rbohD* plants treated with live and heat-killed *Xanthomonas* L148 wild-type and L148::Tn5 mutant strains (OD$_{600}$ = 0.5) (at least 4 independent experiments each with 8 biological replicates, *n* = 24). **d**, **e** Pearson correlation analyses of plant health performance measured as green tissue area against bacterial colonization capacities in the total (**d**) and endophytic (**e**) compartments (R², coefficient of determination) with statistical significance determined through two-sided *t*-test. Results in (**c**, **d**, **e**) were plotted using Locally Estimated Scatter Plot Smoothing (LOESS) with error bars and shadows indicating the standard errors of the mean.

Col-0 with *rbohD* leaves (threshold: *q*-values < 0.05): 563 genes were up-regulated and 2474 genes were down-regulated in Col-0 compared with *rbohD* plants, indicating global bacterial transcriptomic reconfiguration upon leaf colonization as influenced by *RBOHD* (Fig. 7a and c, See Supplementary Data 3 for the details on DEGs). Differentially expressed genes in Col-0 compared with *rbohD* were captured as

candidate genes in the Tn5 mutant screening (73 down-regulated and 29 up-regulated out of 214 candidates), but these candidate genes were rather under-represented (hypergeometric test, ***p*-value = 1.00E-10). Most T2SS apparatus genes including *gspE* were down-regulated in Col-0 as compared to *rbohD* (Fig. 7c–e). Six *Xanthomonas* L148::Tn5 mutants have an insertion in genes annotated as CAZymes,

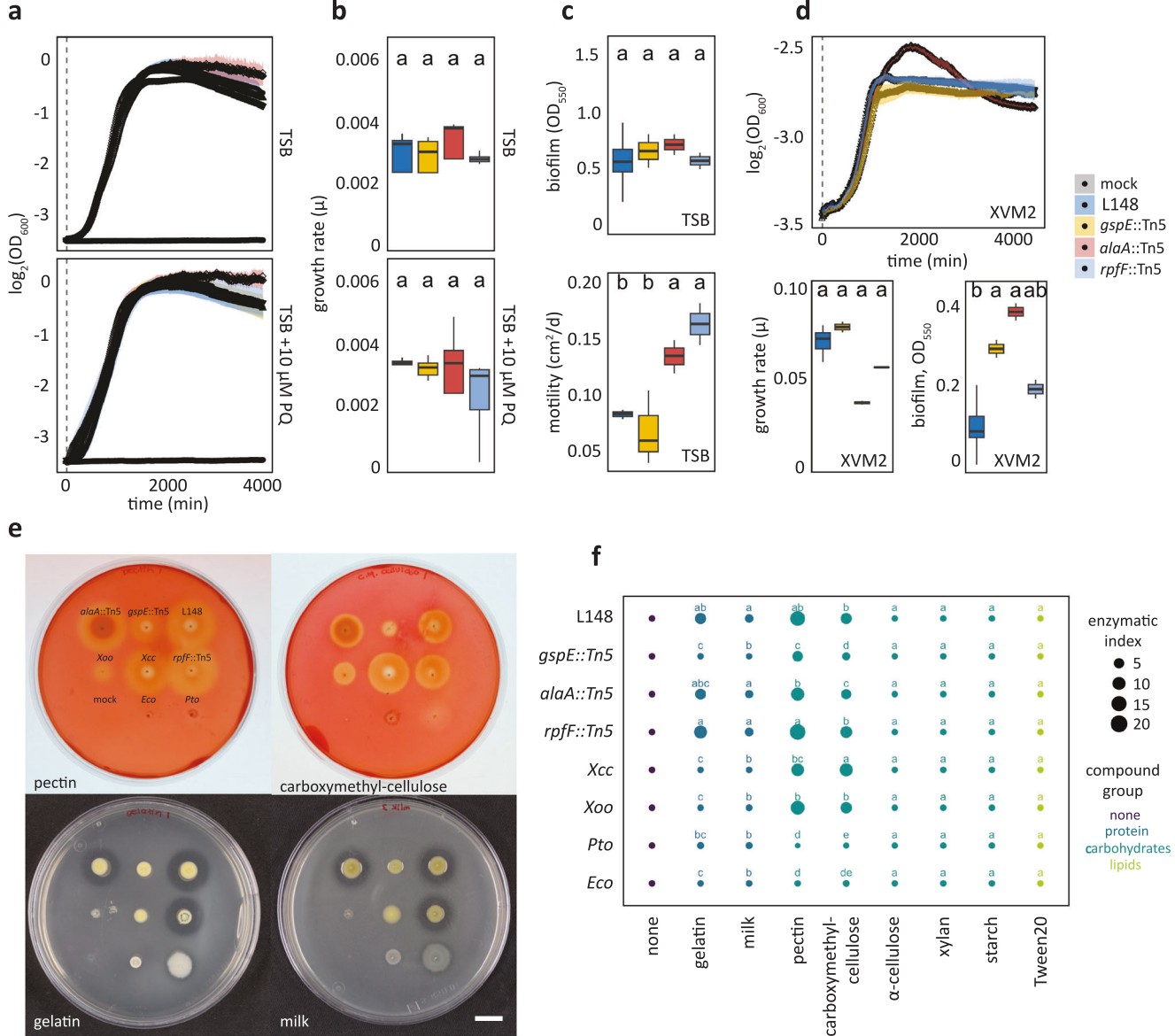

**Fig. 5 | The *Xanthomonas* L148 *gspE*::Tn5 mutant exhibits compromised extracellular secretion activity. a, b** Growth curves (**a**) and rates (**b**) of *Xanthomonas* L148::Tn5 candidate mutant strains in TSB upon chronic exposure to 0 or 10 μM PQ for 4000 min (2 independent experiments each with 3 biological replicates). **c** Biofilm production and motility of *Xanthomonas* L148::Tn5 candidate mutant strains in TSB medium (2 independent experiments each with 2–3 biological replicates, *n* = 6). **d** Growth curves, growth rates, and biofilm production of *Xanthomonas* L148::Tn5 candidate mutants in XVM2 (2 independent experiments each with 2–3 biological replicates, *n* = 6). **e** Exemplary images of plate assays for secretion activities of bacterial strains (L148 = wildtype *Xanthomonas* L148; *gspE*::Tn5; *alaA*::Tn5; *rpfF*::Tn5; *Xcc* = *Xanthomonas campestris* pv. *campestris*; *Xoo* = *X. oryzae* pv. *oryzae*; *Pto* = *P. syringae* pv. *tomato* DC3000; *Eco* = *E. coli* HB101; and mock) for the carbohydrates pectin and carboxymethylcellulose, and gelatin and milk proteins. **f** Enzymatic indices for bacterial strains grown on TSB

supplemented with 0.1% substrates (proteins: gelatin and milk; carbohydrates: pectin, carboxymethyl-cellulose, α-cellulose, xylan, and starch; lipids: Tween20) after 2 day-incubation at 28 °C (3 biological replicates). The enzymatic indices were calculated by subtracting the size of the colony from the zone of clearance, indicative of substrate degradation by the strain after 2–3 days. **b**–**d** the growth rate, μ, was calculated by running rolling regression with a window of 5 h along the growth curves to determine the maximum slope. **b**–**f** Different letters indicate statistically significant differences (two-sided ANOVA with *post hoc* Tukey's test, $P \le 0.05$). Results in (**b**–**d**) are depicted as box plots with the boxes spanning the interquartile range (IQR, 25th to 75th percentiles), the mid-line indicates the median, and the whiskers cover the minimum and maximum values not extending beyond 1.5x of the IQR. Results in (**a**–**d**) are shown as line graphs using Locally Estimated Scatter Plot Smoothing (LOESS) with error bars and shadows indicating the standard errors of the mean.

five of which are significantly down-regulated in Col-0 as compared with *rbohD* inoculated plants. The significantly down-regulated CAZymes in Col-0 plants can potentially degrade plant cell wall components cellulose, pectin, α-glucan, β-glucan, and β-mannan (Fig. 7c, Supplementary Fig. S9). Pathway enrichment analysis revealed that upregulated gene clusters such as clusters 3, 9, and 14 are enriched for biological functions related to chemotaxis and attachment (K15125, K13924, and K05874), while gene clusters down-regulated in Col-0 such as clusters 8, 10, and 12 are enriched for pathways involved in transport and detoxification processes (K02014 and K00799, Fig. 7f, See Supplementary Data 4 for the clustering membership and the enriched GO terms).

Upon closer inspection, expression of the identified candidate genes *gspE* and *alaA* was strongly repressed while *rpfF* was marginally downregulated in Col-0 compared to *rbohD*, which supports the hypothesis that these genes are required and thus regulated in

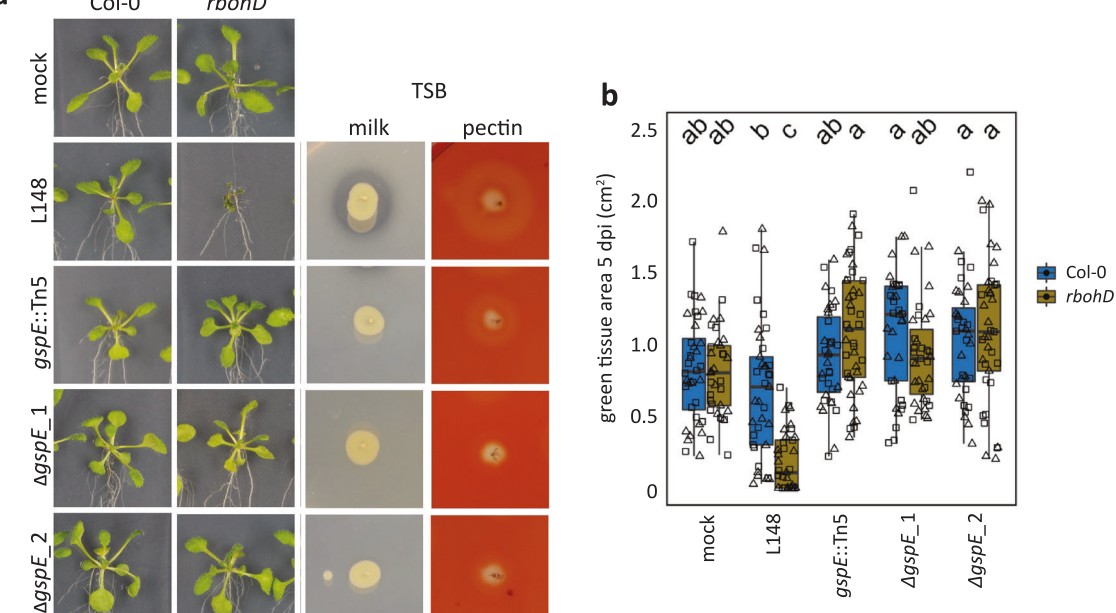

**Fig. 6 | The T2SS component *gspE* is a genetic determinant for the loss of the *rbohd*-dependent pathogenicity of *Xanthomonas* L148. a** Images of axenic Col-0 and *rbohD* plants flood-inoculated (OD$_{600}$ = 0.005) with the wildtype *Xanthomonas* L148, *gspE*::Tn5 mutant and 2 *ΔgspE* lines at 5 dpi. Plate images of secretion activities of the bacterial strains grown on TSB with either milk or pectin as substrate after 2–3 days. **b** Measured green tissue area as the plant health parameter for Col-0 and *rbohD* plants flood-inoculated with the bacterial strains at 5 dpi (2 independent experiments each with 3–5 biological replicates, *n* = 40 and 45, for mock and bacterial strains, respectively). Different letters indicate statistically significant differences (two-sided ANOVA with *post hoc* Tukey's test, *P* ≤ 0.05). Results in (**b**) are depicted as box plots with the boxes spanning the interquartile range (IQR, 25th to 75th percentiles), the mid-line indicates the median, and the whiskers cover the minimum and maximum values not extending beyond 1.5x of the IQR.

immunocompetent wild-type Col-0 plants to prevent disease progression (Fig. 7g). These findings were re-confirmed in independent experiments using qRT-PCR where all the candidate genes were suppressed in Col-0 compared to *rbohD* plants (Fig. 7h). It can be postulated that ROS directly regulates the expression of these genes. Therefore, *Xanthomonas* L148 bacterial cells were grown in vitro in the presence of PQ followed by gene expression analysis. We found that the expression of the candidate genes *gspE, alaA*, and *rpfF* is suppressed in *Xanthomonas* L148 upon chronic exposure to ROS (Fig. 7i). Taken together, these findings suggest that *Xanthomonas* L148 colonization triggers RBOHD-mediated ROS production, which directly inhibits the expression of genes related to virulence, in particular components of T2SS on Col-0 plants. By contrast, the absence of ROS production in *rbohD* mutant plants switches on the pathogenicity of *Xanthomonas* L148, leading to disease onset.

**RBOHD-mediated ROS turns *Xanthomonas* L148 into a protective bacterium**

The phyllosphere microbiota are known to confer protection against foliar pathogens[32] and thus even a conditionally pathogenic microbiota member may provide beneficial services to its plant host. To address this question, Col-0 and *rbohD* plants were pre-colonized with wild-type *Xanthomonas* L148 or *gspE*::Tn5 mutant strain for 5 days and were then challenged with the bacterial pathogen *Pto*. Bacterial titers of *Xanthomonas* L148 and *Pto* were determined for the endophytic and total leaf compartments at 0 and 3 dpi. As *Xanthomonas* L148 killed *rbohD* mutant plants, we were not able to measure *Pto* titers under this condition. Pre-colonized Col-0 plants with either the wild-type *Xanthomonas* L148 or *gspE*::Tn5 mutant had increased resistance against *Pto* (Fig. 8a–c). Interestingly, *rbohD* mutant plants pre-colonized with *gspE*::Tn5 strain showed increased resistance against *Pto* (Fig. 8a, c). Further, Col-0 and *rbohD* plants pre-colonized with *gspE*::Tn5 had slightly better plant performance than the non-inoculated plants after

*Pto* challenge (Supplementary Fig. S10a, b). This also indicates that the protective role of *Xanthomonas* L148 is genetically uncoupled from the *gspE*-dependent pathogenic potential. Invasion by *Pto* did not result in a significant decline in *Xanthomonas* L148 and *gspE*::Tn5 populations (Fig. 8b), indicating a strong colonization competence and resistance of the commensal *Xanthomonas* L148 against pathogen invasion. In summary, these results suggest that RBOHD-produced ROS turns the potentially harmful *Xanthomonas* L148 into a protective bacterium against aggressive pathogen colonization.

## Discussion

Despite extensive studies on how plants recognize microbes and transduce signals within the plant, how immune outputs control the growth and behavior of microbes is still largely unknown. Furthermore, we mostly lack a mechanistic explanation for why certain microbes are sensitive to particular immune responses. In this study, we have investigated the impact of the RBOHD-mediated ROS burst as an early immune output on 20 taxonomically diverse bacteria isolated from healthy *A. thaliana* plants and demonstrated the poor association between RBOHD-mediated ROS burst and bacterial colonization (Supplementary Fig. S4). These findings suggest that *Arabidopsis* recognizes commensal bacteria and produces ROS that does not have a detectable impact on most commensal bacterial colonization. This highlights the notion that the perception of the microbial signal, followed by the cascade of immune signals, and immune execution leading to the restriction of microbial colonization are distinct events. This corroborates our previous finding that plant and transcriptome responses of commensal bacteria are largely uncoupled during an early stage of infection[26]. Plant responses to MAMPs do not necessarily affect the consequence of plant-microbe interactions. This makes sense because MAMPs derived from pathogenic or mutualistic bacterium can induce the same PTI responses[33,34]. Also, different types of MAMPs or its variants can elicit quantitatively different intensities of

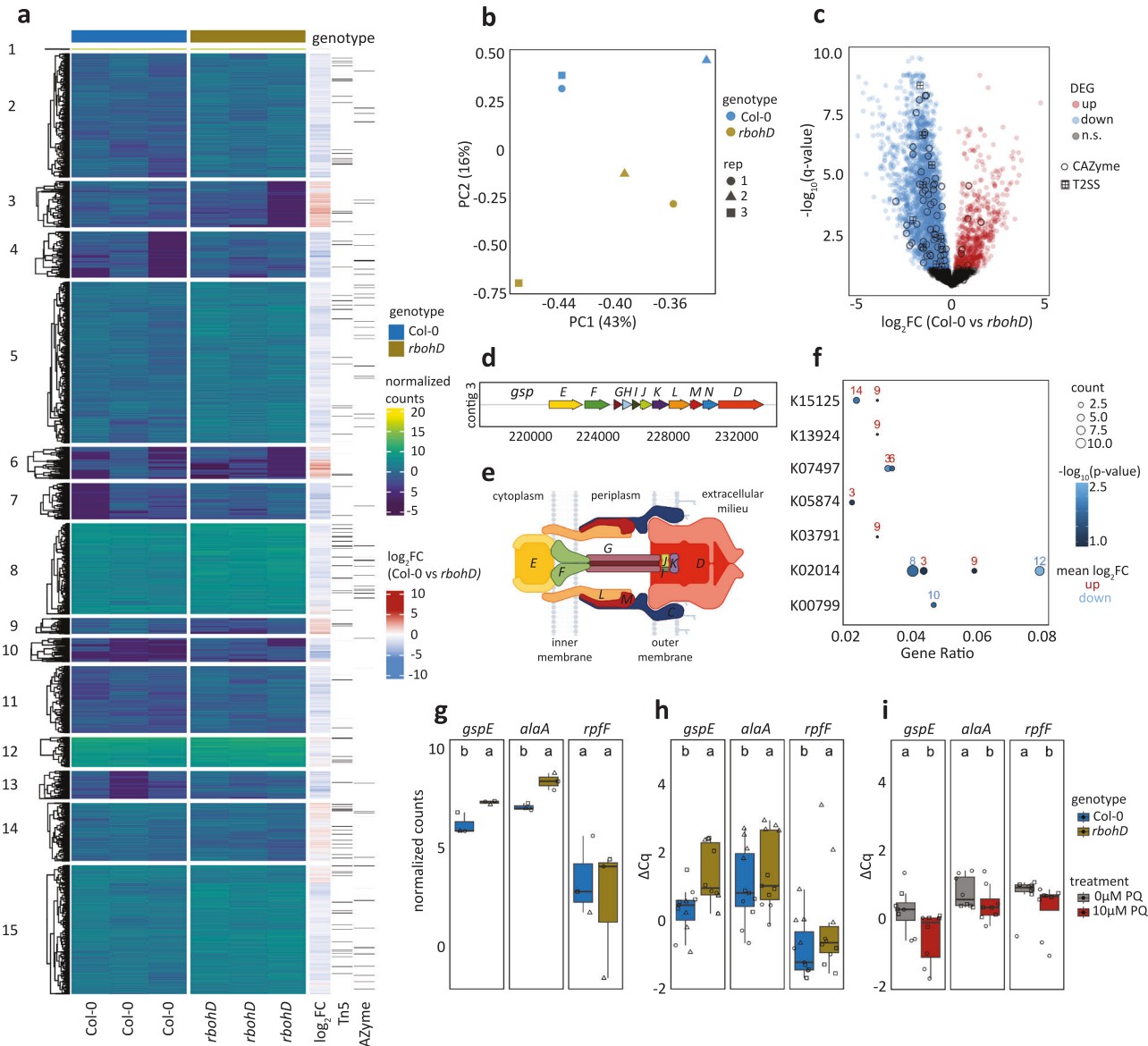

**Fig. 7 | Plant ROS suppresses T2SS genes including *gspE* of *Xanthomonas* L148.**
**a** Heatmap representation of *in planta* bacterial transcriptome landscape of the wildtype *Xanthomonas* L148 in Col-0 and *rbohD* plants. Leaves of 2-week-old plants were flood-inoculated with L148 and samples were taken at 2 dpi. Gene clusters were based on k-means clustering of the normalized read counts. DEGs were defined based on *q*-value < 0.05. Sidebars indicate the log₂ fold changes of Col-0 compared with *rbohD*, *Xanthomonas* L148::Tn5 candidate genes (the 214 candidates), and the genes annotated as CAZyme. **b** Principal component (PC) analysis of the *in planta Xanthomonas* L148 transcriptome for DEGs in Col-0 and *rbohD* plants. **c** Volcano plot of the DEGs with which CAZymes and T2SS component genes are highlighted. **d** Genomic architecture of the T2SS genes. **e** Graphical representation of T2SS assembly. **f** KEGG pathway enrichment analysis (hypergeometric test, *P* ≤ 0.05) of the gene clusters (indicated in numbers) in (**a**). **g** RNA-Seq normalized

counts of *gspE*, *alaA*, and *rpfF*. **h** Independent qRT-PCR experiments for *in planta* expression profiling of *gspE*, *alaA*, and *rpfF*. Experiments were performed as in RNA-seq with 2 independent experiments each with 3–4 biological replicates (*n* = 11, 10, 11, 11, 10, 10, from left to right). **i** qRT-PCR in vitro expression profiling of *gspE*, *alaA*, and *rpfF* in *Xanthomonas* L148 wildtype strain grown in TSB ± 10 µM PQ for 24 h (2 independent experiments each with 3–4 biological replicates, *n* = 8, 8, 8, 8, 7, 8, from left to right). **h**, **i** Gene expression was normalized against the housekeeping gene *gyrA*. Different letters in (**g**–**i**) indicate statistically significant differences (two-sided ANOVA with *post hoc* Tukey's test, *P* ≤ 0.05). Results in (**g**–**i**) are depicted as box plots with the boxes spanning the interquartile range (IQR, 25th to 75th percentiles), the mid-line indicates the median, and the whiskers cover the minimum and maximum values not extending beyond 1.5x of the IQR. Some illustrations were created with BioRender.

immune responses and detection of these molecular patterns might not essentially result in the same immune outputs and different immune readouts are activated by different MAMPs or to the catalog of MAMPs that a particular microbe possesses[35–38]. Thus, the perception of the nuanced compendium of stimuli, triggering the cascade of signals, and the eventual impact of plant immune responses on microbes are tailored for each microbial strain. In this study, we have revealed a mechanism in which RBOHD-mediated ROS changes the growth and behavior of a leaf commensal, *Xanthomonas* L148. This is a

significant advance in our understanding of how plant immune responses manipulate the growth and behavior of its potentially pathogenic microbiota members.

We have demonstrated that plant ROS allows co-habitation with a potentially pathogenic leaf microbiota member, *Xanthomonas* L148 by modulating its behavior. Our results show that the plant host constrains proliferation of this microbiota member by means of ROS as a molecular message to allow cooperation and coexistence. Ecological and reductionist studies have revealed that potentially pathogenic

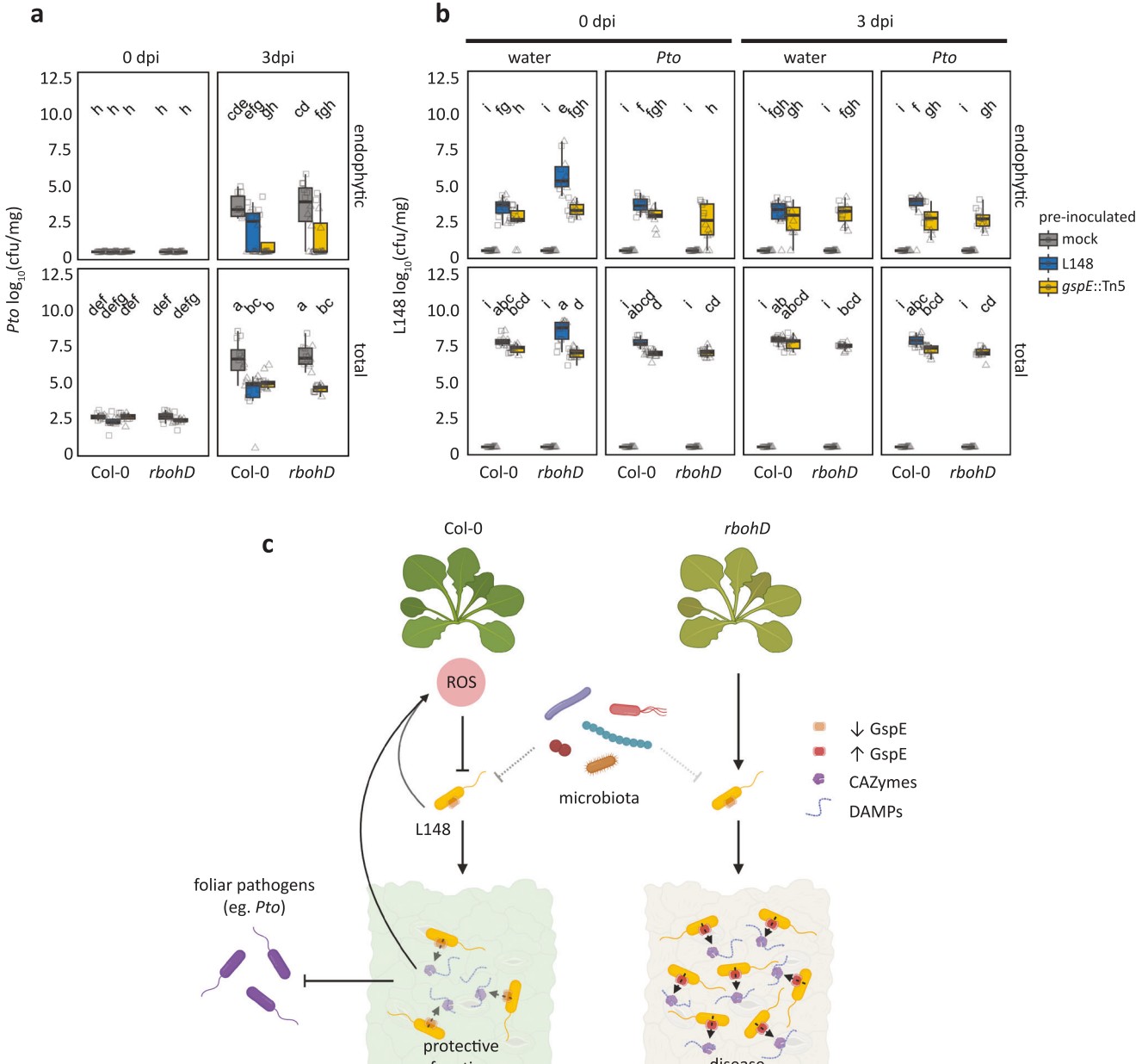

**Fig. 8 | RBOHD-mediated ROS turns *Xanthomonas* L148 into a beneficial bacterium. a, b** 14-day-old Col-0 and *rbohD* plants grown on agar plates were flood-inoculated with wildtype *Xanthomonas* L148 and *gspE*::Tn5 (OD$_{600}$ = 0.005) for 5 days followed by spray infection with *Pto*. Bacterial titers were determined at 0 and 3 dpi (**a**, *Pto*; **b**, L148) (2 independent experiments each with 6 (**a**) or 3–5 (**b**) biological replicates, *n* ≥ 12). Different letters indicate statistically significant differences (two-sided ANOVA with *post hoc* Tukey's test, *P* ≤ 0.05). Results in (**a**, **b**) are depicted as box plots with the boxes spanning the interquartile range (IQR, 25th to 75th percentiles), the mid-line indicates the median, and the whiskers cover the minimum and maximum values not extending beyond 1.5x of the IQR. **c** Mechanistic model for plant ROS licensing of co-habitation with a potentially pathogenic *Xanthomonas* L148 commensal, where the microbe releases MAMPs

that are perceived by plants and trigger ROS production. The T2SS delivers CAZymes to the host to degrade cell wall liberating DAMPs and/or the CAZymes act as a MAMP, which both can potentially bolster ROS generation. The ROS then acts as a molecular beacon for *Xanthomonas* L148 to suppress its pathogenicity, in particular by dampening the activity of T2SS resulting in a negative feedback regulation of the bacterial activity by the plant host. The leaf microbiota weakly affects *Xanthomonas* L148 *in planta* (dashed line). We propose that in wild-type Col-0 plants, the ROS- and the microbiota-mediated suppression of *Xanthomonas* L148 promotes the cooperative behavior of L148 with the host plant and in turn confers protective function against subsequent invasion by foliar pathogens. In the case of *rbohD* mutant plants wherein plant ROS is absent, *Xanthomonas* L148 virulence is unlocked, resulting in disease. Some illustrations were created with BioRender.

strains populate plant hosts without causing disease, and these strains are considered as bona fide constituents of the plant microbiota[29,33,39,40]. Some of these potentially detrimental strains can be deleterious to the host in mono-associations[29,33,39–42]. However, the adverse effects of these potentially pathogenic microbes depend on the host, the environment, and the co-occurring microbes[17,39–45]. It has been shown that simultaneous defects in PTI and the vesicle trafficking pathway under high humidity led to dysbiosis in the phyllosphere and plant disease[11,12]. It appears to be a universal pattern across multicellular organisms that ROS modulates the structure, composition, and function of microbiota. In mice, a decrease in mitochondria-derived ROS is associated with increased gut microbiota diversity[45].

Also, ROS produced via the NOX1 pathway in the colon drives anaerobic growth of *Citrobacter rodentium* and in turn remodel the epithelial milieu[46]. In plants, ROS induces the phytohormone auxin secretion by a beneficial rhizobacterium *Bacillus velezensis* to protect against the damaging effects of plant-derived ROS, allowing efficient root colonization of *B. velezensis*[18]. ROS production in roots constrains *Pseudomonas* establishment in the rhizosphere[16]. It has also been genetically shown that RBOHD-mediated ROS production is integral for maintaining leaf microbiota homeostasis by keeping potentially harmful bacterial members at bay[17]. Nevertheless, the mechanisms by which the plant host selectively constrains potentially pathogenic members of the microbiota and whether these strains are functional to their host remains unclear. Here, through a bacterial genome-wide transposon mutant screen and *in planta* transcriptomics, we have revealed that plant ROS acts as a signaling cue for the potentially pathogenic commensal *Xanthomonas* L148 to suppress its virulence by downregulating its T2SS while promoting its beneficial function. While the massive bacterial transcriptomic landscape reprogramming can be attributed to versatile roles of RBOHD[13] such as cell wall remodeling[47,48], host signaling[49], and stomatal response[50,51], our in vitro experiments support the notion that ROS produced by RBOHD modulates the behavior of the *Xanthomonas* L148.

Other members of the phyllosphere microbiota may partially contribute to attenuating the deleterious effects of *Xanthomonas* L148. However, *RBOHD* is needed for full suppression of L148 deleterious activity in the community context (Supplementary Fig. S6), which is consistent with the observation that *Xanthomonas* L131, a closely-related strain of L148, exerts its detrimental impact on *rbohD* mutant plants in a community context[17]. Closely related, innocuous strains of the plant microbiota out-compete or antagonize its potentially pathogenic counterparts, thereby preventing disease progression but enabling the persistence and co-existence of these strains in nature[23,41]. However, this phenomenon is accession and strain-specific as this commensal-mediated protection is lost in some plant genotypes and a particular harmful *Pseudomonas* strain predominates the microbial community[41]. Thus, allowing potentially pathogenic strains within the microbiota requires stringent control of their function and behavior by host immunity sectors and is facilitated in parts by other members of the plant microbiota.

We have demonstrated that the pathogenicity of *Xanthomonas* L148 depends on the T2SS component *gspE* (Figs. 3d–f and 6a, b). The loss of the killing effect of *gspE*::Tn5 mutants strains on *rbohD* mutant plants can be explained by its compromised secretion activities and hampered colonization of *rbohD* leaves (Figs. 4b, 5e, f, 6a, b, Fig. 7c, and Supplementary Fig. 9a, b). Consistently, the same gene was recently found to be causal for leaf degradation and virulence of *Xanthomonas* L148 in *rbohD* plants, functioning to help releasing substrates for other leaf commensals, consequentially driving the phyllosphere microbiota composition[52]. T2SS is often utilized by plant pathogens to deliver CAZymes which degrade plant cell walls, allowing host invasion and promoting disease[30]. For instance, T2SS allows the root commensal *Dyella japonica* MF79 to efficiently colonize the host and is required for virulence of pathogenic *Dickeya dadantii*[24,53]. These, together with our findings, suggest an important role of T2SS in the establishment of microorganisms in host tissues, making it conceivable that it is targeted by the host to manipulate microbial behavior. Secreted CAZymes could also trigger immune responses such as ROS burst via direct recognition of the CAZyme as a MAMP or release of recognized plant-derived Damage Associated Molecular Patterns (DAMPs) due to their enzymatic action[54–56]. Indeed, we have shown that live T2SS-deficient *gspE*::Tn5 L148 mutant elicited less ROS than wild-type L148, whereas heat-killed wild-type L148 and *gspE*::Tn5 mutant elicited undistinguishable ROS burst, implying that T2SS-mediated CAZyme secretion may further enhance the ROS response (Fig. 4c). Plant ROS might act as a counter-defense of L148 invasion via CAZymes

by dampening T2SS expression (Fig. 7c, Supplementary Fig. 9a, b). The *gspE*::Tn5 mutant had similar leaf endophytic but slightly reduced total colonization in wild-type Col-0 plants compared with wild-type L148. However, the *gspE*::Tn5 mutant did not show increased colonization in *rbohD* in contrast to wild-type L148 (Fig. 4b). This suggests that ROS functions to attenuate T2SS activity albeit incomplete and makes *Xanthomonas* L148 a commensal bacterium in wild-type Col-0 plants. Thus, we propose a model according to which the interaction of *Xanthomonas* L148 and Col-0 plants is based on a delicate balance driven by host ROS levels, resulting in a negative feedback loop to control the potentially pathogenic commensal (Fig. 8c). Our genetic screening and *in planta* bacterial transcriptome analysis (Figs. 3b and 7a; Supplementary Data 2 and 3) also suggest that the conditional pathogenicity of *Xanthomonas* L148 depends on multiple pathways including amino acid metabolism (*alaA*, DAO, *speD*, and *prnA*), quorum sensing and bacterial signaling (*rpfF*, *rsbu_P*), surface adhesion (*sphB1*, *pilD*), and antioxidant metabolism (*ubiB*). Moreover, the plant protective function of *Xanthomonas* L148 against the pathogen *Pto* is not genetically coupled with its *gspE*-dependent pathogenic potential, as the *gspE*::Tn5 mutant can still confer significant resistance against *Pto* in both Col-0 and *rbohD* mutant plants (Fig. 8a, Supplementary Fig. S10a, b). The *gspE*::Tn5 mutant provided better pathogen protection to Col-0 plants than the wild-type *Xanthomonas* L148 despite its lesser leaf colonization, suggesting that efficient colonization does not necessarily translate to pathogen protection. This finding is in line with the recent finding in which a number of microbiota strains that robustly colonize did not confer plant protection against pathogen attack[32]. The mechanisms by which the ROS-tamed *Xanthomonas* L148 confers pathogen protection warrants further investigation. Our finding that RBOHD-mediated ROS targets *Xanthomonas* T2SS provides a mechanism and concept that plant immunity surveils potentially detrimental members of the plant microbiota by suppressing T2SS via ROS.

We have revealed different ROS burst patterns in response to individual members of the plant microbiota that can be categorized into three classes of immune reactivity: immune-active strains can elicit ROS with intact cells; immune-evasive strains only induce ROS when they are heat-killed; and immune-quiescent strains do not elicit ROS whether alive or dead (Fig. 1c, d and Supplementary Fig. S2). Immune-evasive strains can possibly conceal their detection by preventing MAMP release or secrete proteins that degrade/sequester self-derived MAMPs or that target host immune components to suppress immune activation[57].

We have observed that most microbiota members of *A. thaliana* are perceived through the surface-resident PRR EFR (Supplementary Fig. S2), indicating that EF-Tu peptides serve as major bacterial molecules eliciting defense programs in our experimental setup. Consistent with this, a number of strains increased colonization in *efr* mutant plants compared to wild-type Col-0, which emphasizes the fundamental link of microbial perception with bacterial colonization (Supplementary Fig. S3). Our observation also coincides with a GWAS study in which *EFR* was found as a plausible genetic determinant of responses to varying MAMP epitopes in natural populations of *A. thaliana*[58]. Although EFR is a Brassicaceae lineage-specific innovation[7], other EF-Tu fragments seem to be recognized by yet-unknown receptors and are immunogenic to some rice cultivars[59]. Also, inter-family transfer of *A. thaliana* EFR to solanaceous species is sufficient to confer broad-spectrum resistance to pathogens, indicating that components acting downstream of EFR perception are at least in parts evolutionarily conserved[60]. These findings suggest that EF-Tu peptides might be a prevalent microbial motif for host detection in various plant species.

Emerging evidence suggests that plant immunity modulates microbial processes required for virulence in addition to its effects on general microbial metabolism, including protein translation[31,61]. For

instance, the secreted aspartic protease SAP1 inhibits *Pto* growth by cleaving the *Pto* protein MucD in *A. thaliana* leaves[62]. Plants target the iron acquisition system of *Pto* to inhibit *Pto* growth during effector-triggered immunity[31]. The defense phytohormone salicylic acid and the specialized metabolite sulforaphane inhibit the type III secretion system of pathogenic *Pto*[61,63]. Our finding is consistent with the notion that plant immunity targets microbial virulence to allow microbes to cohabit, which can be a better plant strategy than eliminating microbes as plants need to maintain a functional microbiota and potentially harmful microbes can even provide a service to the host.

## Methods

### Plant materials and growth conditions

The *A. thaliana* Col-0 accession was the wild-type and the genetic background of all the mutants utilized in this study. The mutants *fls2*[6] (SAIL_691C4), *efr*[7] (SALK_068675), *cerk1*[9] (GABI_096F09), *fec*[11], *bbc*[11], and *rbohD*[13] (atrbohD D3) were previously described. For agar plate assays, seeds were sterilized with Cl2 gas for 2 h[64]. Seeds were then stratified for 2–3 days at 4 °C on 0.5 × Murashige and Skoogs (MS) medium agar with 1% sucrose, germinated for 5 days, and subsequently transplanted to 0.5 × MS plates without sucrose. Plants were grown in a chamber at 23 °C/23 °C (day/night) with 10 h of light. Then, 14-day-old seedlings were inoculated with bacterial strains and were harvested or phenotyped at the indicated time points. For ROS burst and infiltration patho-assays, plants were grown in greenhouse soil for 5–6 weeks in a chamber at 23 °C/23 °C (day/night) with 10 h of light and 60% relative humidity (See Supplementary Data 5 for details of the plant genotypes used).

### Bacterial strains and growth conditions

All the bacterial strains derived from the AtSPHERE were previously described[25]. *Pseudomonas syringae* pv. *tomato* DC3000 (*Pto*) and *Pto* lux were described previously[65,66]. All bacterial strains were grown in 0.5x Tryptic Soy Broth (TSB) for 24 h, harvested through centrifugation, washed twice with sterile water, and diluted to the appropriate OD600 (See Supplementary Data 1 for the list of bacterial strains used).

### ROS burst measurement

ROS burst was determined as in Smith and Heese, 2014 with slight modifications[67]. In brief, bacterial strains were grown in TSB at 28 °C for 16–18 h with shaking at 200 rpm. Cells were harvested, washed twice with sterile water, and diluted to OD600 = 0.5 in sterile water. The day before the assay, leaf discs (4 mm) from leaves of the same physiological state and size from 5-to-6-week-old plants grown in a chamber at 23 °C/23 °C (day/night) with 10 h of light were harvested, washed twice with sterile water every 30 min, immersed in sterile water in 96-well plates, and incubated in the same growth chamber for 20 h. Prior to the assay, the elicitation solution was prepared by adding 5 μL 500x horseradish peroxidase (HRP, P6782-10MG, Sigma-Aldrich) and 5 μL 500x luminol (A8511-5G, Sigma-Aldrich) to 2.5 mL of bacterial suspension, 1 μM MAMP solutions (flg22 [ZBiolab inc.], elf18 [Eurofins], chitinDP7 [N-acetylchitoheptaose, GN7, Elicityl]), or sterile nanopure water as mock. During the assay, the water was carefully removed from the 96 well-plate and 100 μL of the elicitation solution was added to the 96-well plate. With minimal delay, the luminescence readings were obtained for 60 min using a luminometer (TriStar2 Multimode Reader, Berthold).

### Commensal bacterial colonization assay

To prepare the bacterial inoculum, all bacterial strains were grown in 0.5× TSB for 24 h, harvested through centrifugation, washed twice with sterile water, and suspended in sterile water (final OD600 = 0.005). Two-week-old seedlings grown on 0.5× MS medium agar in a chamber at 23 °C/23 °C (day/night) with 10 h of light were flood-inoculated with these bacterial suspensions (15 mL per plate, incubated for 1 min),

drained, air-dried for 5 min and then incubated in the same growth chamber. Leaf samples were aseptically harvested at 3 to 5 dpi, weighed, and plated for two compartments: for the total compartment, leaves were directly homogenized in 10 mM MgCl2 with a homogenizer (TissueLyser III, Qiagen), serially diluted with 10 mM MgCl2, and plated on 0.5 × TSB agar; for the endophytic compartment, leaves were surface-sterilized with 70% ethanol for 1 min, washed twice with sterile water, homogenized, serially diluted, and then plated as for the total compartment. Colonies were allowed to grow at 28 °C, and photographs were taken for 1–3 days. Colonization was expressed as cfu mg$^{-1}$ sample.

### Generation of bacterial mutants

A *Xanthomonas* L148::Tn5 library was constructed via conjugation of *Xanthomonas* L148 with *E. coli* SM10λpir harboring puTn5TmKm2[68] in which both strains were mixed in equal portions (OD600 = 0.10), spot-plated on TSB medium, and incubated for 2 d at 28 °C. The resulting mating plaques were diluted and plated on TSB with kanamycin and nitrofurantoin for selection for L148 transformants and counter-selection against *E. coli*, respectively. To constitute the entire library, around 7000 individual colonies were picked, re-grown in 0.5 × TSB, aliquoted for glycerol stocks, and stored at −80 °C. Around 20 strains from this *Xanthomonas* L148::Tn5 library were randomly selected for confirmation of Tn5 insertion in the genome via nested PCR (first PCR with primers FDE117 and FDE118; second PCR with primers FDE119 and mTn5AC) and the final amplicons were Sanger-sequenced (see Supplementary Data 6 for details). For the generation of targeted deletion mutants for *gspE*, the pK18mobsacB suicide plasmid[69] (GenBank accession: FJ437239) was PCR linearized (primers FDE234 and FDE235) with Phusion Taq polymerase (F-5305, Thermo Scientific); 750 bp of upstream (primers FDE278 and FDE279) and downstream (primers FDE280 and FDE281) flanking regions of *gspE* coding sequence with terminal sequences overlapping with the linearized pK18mobsacB were amplified using Phusion Taq polymerase (F-5305, Thermo-Scientific) and were sequence-verified. The plasmid construct was assembled using Gibson cloning[70] (E5510S, New England Biolabs) following the manufacturer's instructions. The plasmid construct was transformed into *E. coli* cells (DH5α strain) and then delivered into *Xanthomonas* L148 via triparental mating with the helper strain pRK600[71]. Transformants were selected using kanamycin and nitrofurantoin and the second homologous recombination was induced with sucrose in 0.5× TSB. The deletion mutants were individually picked and stored at −80 °C in glycerol stocks and were verified by PCRs (using primers FDE196 and FDE197 for the presence of the plasmid with the inserts; primers FDE125 and FDE126 for the presence of *gspE* gene in the genome; and primers FDE279 and FDE280 for the removal of *gspE* gene in the genome) and Sanger-sequencing, and were plated on 0.5× TSB containing 10 μg/mL kanamycin. True deletion mutants should not contain the plasmid, lose the *gspE* gene, and be sensitive to kanamycin (See Supplementary Data 6 for list of primers and PCR profile used).

### L148::Tn5 library 96-well screening

Seedlings of *rbohD* were grown in 96-well plates with 500 μL 0.5× MS agar with 1% sucrose for each well and incubated in a growth chamber at 23 °C/23 °C (day/night) with 10 h of light for 14 days. Concomitantly, the Tn5 insertion mutants (~7000 individually picked colonies) were grown in 96-well plates with TSB at 28 °C for 3 days with 200 rpm agitation till saturation. The resulting bacterial suspension was diluted six times (resulting in a concentration of ~6 × 10$^9$ bacterial cells per mL) and 20 μL aliquots were inoculated onto the seedlings. Plants were phenotyped for survival after 5 days. The resulting 214 *Xanthomonas* L148::Tn5 candidate strains which showed the loss of the *rbohD* killing activity from the two independent 96-well plate screenings were genotyped to identify the Tn5 insertion locus in the genome via nested

PCR (first PCR with primers FDE117 and FDE118; second PCR with primers FDE119 and mTn5AC) and the final amplicons were Sanger-sequenced (see Supplementary Data 6 for list of primers and PCR profile used and Supplementary Fig. S7). The 124 *Xanthomonas* L148::Tn5 candidate mutants which have insertions on genes with functional annotations (please see Supplementary Data 2 for the list) were further screened using plants grown in agar plates to re-evaluate the phenotypes as described for the commensal bacterial colonization assay.

## In vitro assays

For instantaneous ROS treatment, *Xanthomonas* L148 was grown for 24 h, pelleted, and diluted to $OD_{600} = 0.02$. A 500 µL of the bacterial suspension was mixed with $H_2O_2$ (H10009-500ML, Sigma-Aldrich) at final concentrations of 0–2000 µM, incubated for 30 min, and plated for colony counts. Similarly, 500 µL of the bacterial suspension was mixed with 1 mM xanthine (X7375-10G, Sigma-Aldrich) and 10 U/mL xanthine oxidase from bovine milk (X4875-10UN,Sigma-Aldrich) to generate $O_2^{-1}$, and samples were plated at different time points (1 mol of xanthine is converted to 1 mol $O_2^{-1}$ with 1 U xanthine oxidase at pH 7.5 at 25 °C in a min, thus 0, 2, 4, 10, 20, 40, 60, and 80 min incubations should have produced $O_2^{-1}$ equivalent to 0, 50, 100, 250, 500, 1000, 2000 µM respectively) for colony counts. Chronic exposure to ROS was implemented by growing the strains in TSB ± 10 µM paraquat (856177-1 G, Sigma-Aldrich), a ROS-generating compound, for 3 days while obtaining $OD_{600}$ readings using spectrophotometer (Tecan Infinite Microplate reader M200 Pro) to calculate growth curves and rates. The candidate *Xanthomonas* L148::Tn5 mutants were phenotyped in vitro via growing bacterial culture with an initial inoculum of 10 µL $OD_{600} = 0.1$ in 96-well plates supplemented with 140 µL TSB or XVM2 (a minimal medium designed for *Xanthomonas* strains[72]) for 3 days while obtaining absorbance readings at $OD_{600}$ using a spectrophotometer (Tecan Infinite Microplate reader M200 Pro) to calculate growth curves and rates. The resulting cultures were gently and briefly washed with water and cells adhering on the plates were stained with 0.1% crystal violet (27335.01, Serva) for 15 min. The staining was solubilized with 125 µL 30% acetic acid (A6283, Sigma-Aldrich) to quantify biofilm formation at $OD_{550}$ using a spectrophotometer (Tecan Infinite Microplate reader M200 Pro). Motility was assayed by point-inoculating bacterial cultures ($OD_{600} = 0.1$) on 0.5× TSB with 0.8% agar and colony sizes were measured after 2 to 3 days. Secretion activities were profiled via point-inoculating (1 µL culture, $OD_{600} = 0.1$) bacterial strains on 0.5× TSB agar with 0.1% substrate-of-interest (carbohydrates: pectin, carboxymethyl-cellulose, α-cellulose, xylan, starch; protein: milk and gelatin; lipid: Tween 20), incubated at 28 °C for 2 days. For gelatin, halo of degradation was visualized by incubating the plates in saturated ammonium persulfate for 15 min. For carbohydrates, clearance zones were visualized by staining the plates with 0.1% Congo red (C-6767, SigmaAldrich) for 15 min followed by washing with 6 ppm NaCl solution (0601.1, Roth). All plates were photographed before and after the staining procedures. The enzymatic indices were calculated by dividing the zones of clearing by the colony size.

## In planta bacterial RNA-Seq

The *in planta Xanthomonas* L148 RNA-Seq was done in accordance to Nobori et al.[73]. Briefly, two-week-old plants grown in agar plates were flood-inoculated with *Xanthomonas* L148 ($OD_{600} = 0.005$ in 10 mM $MgCl_2$) and shoots of ~150 plants were harvested and pooled per sample at 2 dpi when bacterial populations were similar between Col-0 and *rbohD* plants. Samples were harvested, snap-frozen in liquid $N_2$, and stored at -80 °C until RNA extraction. The whole experiment was repeated three times. Samples were crushed with metal beads and incubated for 24 h at 4 °C with the isolation buffer[73]. Bacterial cells were separated from the plant tissue through 6 µm filter mesh and centrifugation at 1300 g for 10 s. The RNA was isolated from the

bacterial pellets using TRIzol (15596026, Invitrogen) and were treated with Turbo DNase (AM1907, Invitrogen) prior to sending to the Max Planck-Genome-Centre Cologne for RNA Sequencing with plant ribo-depletion and cDNA library construction (Universal Prokaryotic RNA-Seq Library Preparation Kit, Tecan) using the Illumina HiSeq 3000 system with 150 bp strand-specific single-end reads resulting in ~10 million reads per sample. The resulting reads were mapped to the *Xanthomonas* L148 genome[25] using the align() function with the default parameters in Rsubread package[74] to generate BAM files. Mapping rates ranged from 20–46%, which is within the expected values[32]. Mapped reads were counted using DESeq2[75] using the function featureCounts() from the BAM files and were normalized using the voom() function in limma package[76] prior to analysis. RNA-Seq raw reads and processed data were deposited in the NCBI GEO repository with accession number GSE226583.

Upon passing quality checks (assessing batch effects through PCA and MA plots for data dispersion), differentially expressed genes were determined using a linear model (gene expression ~0 + genotype + rep; contrast = Col-0 - *rbohD*) and Empirical Bayes statistics with eBayes() function in limma[76]. False discovery rates were accounted for p-values using qvalue[77]. The threshold for significantly differentially expressed genes was set to *q*-value < 0.05. Principal component analysis was done using the prcomp function[78]; the optimal number of clusters was determined using NbClust() function in NbClust package[79], cluster memberships were computed with the k-means algorithm[80], heatmaps were generated using Heatmap() function in ComplexHeatmap package[81], and pathway enrichment analysis was done for each of the identified gene clusters using enricher() function in clusterProfiler package in R[82].

## Synthetic community experiment

Two-week-old plants grown in agar plates in a chamber at 23 °C/23 °C (day/night) with 10 h of light were flood-inoculated with *Xanthomonas* L148 with or without the leaf-derived synthetic communities (LeafSC, 9 leaf prevalent and functional leaf isolates[27–29]) in two different doses: $L148_{P1}$ + LeafSC contains equal portions of each strain including L148 in the inoculum (Xanthomonas L148/LeafSC, 1:9, each strain would have a final $OD_{600} = 0.01$ totaling to $OD_{600} = 0.09$ for LeafSC) and $L148_{P9}$ + Leaf SC contains a population of *Xanthomonas* L148 that equals the entire bacterial load of the LeafSC (Xanthomonas L148/LeafSC, 9:9, L148 and the LeafSC at $OD_{600} = 0.09$), and were incubated in the same growth chamber. Plants were phenotyped for shoot fresh weights at 14 dpi (See Supplementary Data 1 for list of bacterial strains).

## Protective function experiment

Two-week-old plants grown in agar plates in a chamber at 23 °C/23 °C (day/night) with 10 h of light were flood-inoculated with *Xanthomonas* L148 strains ($OD_{600} = 0.005$) and incubated for 5 days. *Pto* lux[66] (OD = 0.005) or water was aseptically spray-inoculated (~200 µL per plate) onto the pre-colonized plants. Samples were collected at 0 and 3 dpi to count L148 and *Pto* colonies for different leaf compartments. For the total compartment, leaves were directly homogenized, serially diluted, and plated; for the endophytic compartment, leaves were surface-sterilized with 70% ethanol for 1 min, washed twice with sterile water, homogenized, serially diluted, and then plated. Colonies were allowed to grow on 0.5× TSB agar at 28 °C, and photographs were taken for 1–3 days. Colonies of the *Pto* lux[66] were differentiated against *Xanthomonas* L148 strains via their color and luminescence, and colonization was expressed as cfu $mg^{-1}$ leaf sample.

## qPCR analysis

Bacterial RNA was isolated from plant samples inoculated with *Xanthomonas* L148 2 dpi or from bacterial pellets from *Xanthomonas* L148 grown in 0.5x TSB with or without 10 µM PQ using TRIzol (15596026, Invitrogen) followed by treatment with Turbo DNase (AM1907,

Invitrogen). The cDNA libraries were synthesized with 1 μg RNA input using SuperScript II reverse transcriptase (18064-014, Invitrogen) and random hexamers as primers following the manufacturer's instructions. An input of 50 ng of cDNA was used for qPCR analyses (CFX Connect Real-Time System, Biorad) of the bacterial genes (please see Supplementary Data 6 for the list of primers and genes tested). The ΔCq was computed by subtracting the Cq of the gene-of-interest from the Cq of the *gyrA* gene from *Xanthomonas* L148.

## Statistical analysis

The R programming environment (R version 4.2.2) was used for data analysis and visualization[78]. No statistical test was implemented to predetermine the sample size. The experiments were done in a randomized manner wherein plants were shuffled weekly inside the growth chambers during the course of the experiment. The investigators were not blinded due to the strong *rbohD* plant phenotype in response to *Xanthomonas* L148. The data were inspected for the assumptions of the linear model (homoscedasticity, independence, and normality) and were normalized, if necessary, prior to statistical analysis using two-sided ANOVA with *post hoc* Tukey's HSD test or the Least Significant Difference (LSD) test for multiple comparisons using the package agricolae[83]. No data was excluded from the analyses.

## Genomic interrogation for CAZyme functions

Genomes for *Xanthomonas* L148 and other Xanthomonadales strains within the AtSPHERE[25] and known *Xanthomonas* pathogens (downloaded from NCBI; Sayers, et al.) were annotated for CAZyme functions (http://www.cazy.org/)[84] using the eggnog mapper (http://eggnog-mapper.embl.de/)[85] to determine the CAZyme repertoire of the bacterial strains and their potential substrates[86].

## Reporting summary

Further information on research design is available in the Nature Portfolio Reporting Summary linked to this article.

## Data availability

The *in planta* bacterial RNA-Seq data reported in this paper have been deposited in the Gene Expression Omnibus (GEO) database with the accession number GSE226583. Source data are provided with this paper.

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

## Acknowledgements

We thank Neysan Donnelly for editing and Wanqing Jiang for providing helpful comments on the manuscript. This work was supported by the National Natural Science Foundation of China (32250710139 to K.T.), the National Key R&D Program of China (2022YFA1304403 to K.T.), HZAU-AGIS Cooperation Fund (SZYJY2021007 to K.T. and X.H.), the Max Planck Society (to P.S.-L and K.T.), and a German Research Foundation (DFG) grant (SPP2125) (to P.S.-L and K.T.).

## Author contributions

F.E. and K.T. conceived the research. F.E., X.H., P.S.-L., and K.T. designed the research. A.M. designed and constructed *Pto* lux. F.E. performed all of the experimental work and the analysis of the data. F.E. and K.T. wrote the manuscript with input from all the authors.

## Funding

## Competing interests

The authors declare no competing interests.
