## [Peer Review File · Nature Communications]

Commensal lifestyle regulated by a negative feedback loop between *Arabidopsis* ROS and the bacterial T2SSREVIEWER COMMENTS

Reviewer #1 (Remarks to the Author):

General assessment

The paper "Commensal lifestyle regulated by a negative feedback loop between Arabidopsis ROS and the bacterial T2SS." discusses the interaction between Arabidopsis and a Xanthomonas strain called L148. The study explores the relationship between reactive oxygen species (ROS) produced by Arabidopsis repress the bacterial type II secretion system (T2SS) of a commensal Xanthomonas isolate. The authors propose that there is a negative feedback loop between Arabidopsis ROS and the T2SS, which helps regulate the commensal lifestyle of the bacteria on plant leaves. The paper is clearly written and the topic interesting, however I do have some reservations that in my opinion should prevent publication in Nature communications:

- 1) Novelty. Although the authors do cite Pfeilmeier et al (reference 17), I don't think they give sufficient credit to that paper. This paper already identified rbohD as a very important gene in retaining homeostasis in the leaf microbiota, and showed that especially Xanthomonas leaf148 and leaf131 were suppressed by RBOHD and become pathogens on rbohD mutant plants. I feel the authors of this study claims these discoveries too much as their own.
- 2) The true novelty lies in the mechanism by which L148 kills the immunocompromised rbohD plant. Through a mutant screen the authors identify 214 candidate genes involved in pathogenicity and focus on the T2SS of L148. They show L148 growth is insensitive to ROS, but that T2SS is suppressed in WT and not in rbohD mutant plants. However a large amount of bacterial genes is differential expressed between the two plant genotypes (almost 75%) . This suggests that the ROS do have very brought spectrum effects and that L148 is not so insensitive to ROS.
- 3) Related, unless there is a typ-O, I think the claim that "DEGs significantly enriched for the candidate genes detected from the Xanthomonas L148::Tn5 mutant screening" (line 318) is not true. 2946 DEGs out of a total of 4,208 genes (according to genbank) in the L148 is almost 75% of the genes of L148. That would mean that you would expect to find more than 75 % of the 214 candidate genes from the mutant screen if there is over representation and 102 genes is a significant underrepresentation. I found in general the applied statistics were opportunistic and sloppy.
- 4) Moreover, T2SS are very common among gram negative bacteria and it is widely demonstrated that T2SS mediated secretion of extracellular enzymes is important for bacteria in natural environments. It seems to me that, even though the gspE::tn5 (T2SS) mutant was not impaired in its growth in liquid growth medium, it might still very well be growth impaired in planta. I'm having a hard time imaging that the T2SS is what makes this Xanthomonas specifically pathogenic to rbohD mutant plant. My conclusion would thus be yes the T2SS is required for full virulence of L148 as L148 cannot grow properly without it, but this not likely the reason why L148 has this effect on rbohD mutant plant and I don't think this paper is (line 390) "is a significant advance in our understanding of how plant immune responses manipulate bacterial growth and behavior".

Detailed comments

Line 27-30: We found that commensal bacteria isolated from healthy Arabidopsis plants trigger diverse patterns of reactive oxygen species (ROS) production via the NADPH oxidase RBOHD that selectively inhibited specific commensals, notably Xanthomonas L148. - ->this is not something the authors found, it was previously found by: <https://doi.org/10.1038/s41564-021-00929-5>

Line 34: In planta bacterial transcriptomics revealed that RBOHD suppresses most T2SS gene expression including gspE. - ->is this direct effect on T2SS or does it affect general metabolic activity?

Line 74 ROS also prohibits dysbiosis in A. thaliana leaves by suppressing Xanthomonas17- ->Xanthomonas is a bit simplistic. This paper showed that for the actual Xanthomonas strain that is also investigated in this study is kept in check, preventing dysbiosis. It appears the authors try to sell that an original research find..

Line 94o (T2SS) has been shown to be necessary for the pathogenesis of many 95 phytopathogenic bacteria and mainly functions to secrete enzymes to degrade host 96 barriers and promote virulence20 - ->the function of the t2ss is more general

Line 161-165. It should not come as a surprise to the authors that RBOHD suppresses L148, this was shown before and reported in Nature microbiology

Line 224 “ we developed a protocol” - ->This reads like a simple adaptation of the protocol developed by Pfeilmeier et al.

Line 233 only 124 had transposon insertions in genes with functional annotations. - ->this is surprising! Why is this? Did many genes in the genomes miss an annotation or did you enrich for

Figure 4a-b. it is a bit weird to test this with ANOVA, you would want to know whether the mutant is affected in its ability to colonize and cause disease in *robhd* plants. A simple t-test between wildtype and *robhd* would be better and much more sensitive than an ANOVA with so many comparisons. Line 251 it looks to me that *gspE::tn5* is impaired in its colonization, but the ANOVA is not sensitive enough to capture this as a result of the many comparisons. In any case it is ridiculous to do the ANOVA over both time points

Lines 301-304: *Xanthomonas* L148 pathogenicity is exerted in the absence of ROS through RBOHD while our in vitro results do not indicate general cellular toxicity of ROS. Thus, it can be assumed that RBOHD-mediated ROS production suppresses virulence of *Xanthomonas* L148. - ->is this true? 2,474 downregulated genes points towards something massive experienced by the *Xanthomonas* in wildtype that is missing in *robhd*. Besides RBOHD mediates ROS production but likely many other things in the plant.

Lines 317 Strikingly, most T2SS apparatus genes including *gspE* were down-regulated in Col-0 as compared to *robhd*. - ->This is not 'striking' considering more than half the genome is downregulated

Line 318-302 I either don't understand or have trouble believing that “DEGs were significantly enriched for the candidate genes detected from the *Xanthomonas* L148::Tn5 mutant screening.” According to genbank *Xanthomonas* L148 has 4,208 genes). 102 genes overlapping between the 214 candidate genes (almost half of the candidate genes) from the mutant screen and 2946 DEGs (3/4 of the total genes) would be a significant UNDER representation of the candidate genes (<https://systems.crump.ucla.edu/hypergeometric/index.php>)

Fig. 8 Like fig 4, also here all treatments are compared to all by ANOVA, which reduces the sensitivity of the statistical test a lot . In this way, the authors can claim that “Interestingly, *robhd* mutant plants pre-colonized with *gspE::Tn5* strain showed increased resistance against *Pto*, resembling *Xanthomonas* L148 pre-colonized Col-0 plants (Figure 8a, c).” However, there is a clear difference in number of *Pto* CFUs between wild-type and mutant leaf148 in fig 8a, that would have been significant if the authors would have chosen the statistical test less opportunistically.

Line 353-373 these experiments were all performed on axenic plants. *Xanthomonas* leaf148 does not kill Col-0 plants but does colonize it. Hence, here we were looking here at the invasive success of *Pto* on its host with a competition and comparing to the success without it. Although I find these results intriguing I refrain from calling it a beneficial bacterium. In this artificial situation, my expectation is that any commensal bacterium would benefit the plant by competitive exclusion of a pathogen and suggest that the authors prove it can protect a plant growing in natural soil if they want to call it a plant-beneficial bacterium.

Lines 392-394 “We have demonstrated that plant ROS licenses co-habitation with a potentially detrimental *Xanthomonas* L148 while it trains to guard against aggressive leaf pathogens.” In line with my previous comment I think this is a bridge too far. Yes ROS keep the commensal in check and prevents it from becoming a pathogen and yes a plant colonized by a commensal will be less easily invaded than an axenic plant, but ROS do not “train L148 to guard against”.

Reviewer #2 (Remarks to the Author):

In the paper by Entila et al. a mechanism of modulation of the pathogenic phenotype of an otherwise commensal microbe *Xanthomonas* L148 by ROS production via RBOHD in *Arabidopsis* is discovered. This work does a significant amount of screening of ROS production and colonization for 20 strains in combination with various MAMPs across various immune compromised *Arabidopsis* genotypes. This uncovers that strain L148 is significantly enriched in the *rbohd* genotype deficient in ROS production and causes a pathogenic phenotype not observed in the Col-0 plants where ROS production via RBOHD is intact. Then the authors identify the genetic factors determining this phenotype in L148. Through transposon mutagenesis they identify hundreds of L148 mutant strains with a loss of the pathogenic phenotype on *rbohd* plants. They identify three strains in particular with insertions in *gspE*, *alaA*, and *rpfF* and show their phenotypes (biofilm formation, motility, CAZyme production, etc.). The authors focus on the *gspE* mutant to show that there is a loss of CAZyme secretion but not a loss in colonization. Using transcriptomics in L148 they show that many of the *gsp* and CAZyme genes are downregulated in Col-0 vs the *rbohd* genotype suggesting these genes are responsive to ROS production in the Col-0 genotype. This is developed into a model where ROS production regulates the *gsp* system in L148 and this prevents it from becoming pathogenic on plants.

Major comments:

I think much of the language to describe the interaction between the plant and L148 suggests the plant is very active in the interaction to prevent L148 from becoming a pathogen.
at line 105 “converting *Xanthomonas* L148 into a commensal”. “Converting”
line 339-340 “thus tightly regulated by immunocompetent wild-type Col-0 plants to prevent disease progression”. “tightly regulated by”
line 392-393 “licenses” and “trains”
line 394: “the plant host constrains... for its own benefits”
line 414: “the plant host selectively constrains”
line 462: “targeted by the host”

The default state in nature is that the wild type plant is going to produce ROS. It is possible that the microbe is evolved to not activate its pathogenic secretion of CAZymes in this situation so that it can survive mutualistically with the plant. Much of this language should likely be reconsidered to discuss the coevolution of this interaction and the mutualistic balance. I think the authors capture this best in lines 448-453. This description makes both the plant and bacteria less active in controlling the other.

Figure 5 is a nice collection of data, but it seems less central to the main story. Could parts or all of this be moved to supplemental information?

In Figure S8 and approximately line 282: How was it decided which strains are pathogens? In particular how do we know that the Xma strain from the human microbiome is not a plant pathogen?

Supplementary figure 10 and the section starting at line 353. The interpretation of this data seems to go beyond what it shows. What happens if you pre-colonize with other commensals and then add Pto? Is this exclusion of Pto exclusive to L148 or would any other commensals fill this niche and exclude Pto? The description that L148 “promotes plant fitness in the presence of pathogens” (line 367) and that ROS turns L148 “into a beneficial bacterium, thereby protecting the plant from aggressive pathogen colonization” (line 372) seem to be giving L148 too much credit when no other strains were tested in this assay.

In the model of Figure 8c. I don't understand the microbiota suppressing L148 at the top center of the model. In figure S6 L148 always has an effect on *rbohd* plants with and without the SynCom. And in Col-0 plants there is no difference between L148 alone and in the SynCom.

In Figure S9. How are the “potential substrates” of these CAZymes determined? Some GH families have very broad substrate activity. For example GH5 can be active on beta-glucan, beta-mannan,

xylan, etc. and is only listed as beta-mannan in the Figure S9a. It may be better to keep this entire figure in the context of the CAZyme family number annotation (GH, PL, CBM) and not suggest the substrates if they are not experimentally validated.

A few points in the methods are missing some critical detail to replicate the assays:

Line 545: The description of how the flood inoculation is performed is not clear. How much volume of the OD=0.005 bacteria solution? How long is it placed in the plate?

Line 588: 96 well plate assay. How much agar in each well? What are the plant growth conditions?

Line 648: How are the bacteria separated from the plant tissue specifically. What centrifugation speed and time?

Line 695: what "chemiluminescence" was used?

Minor comments:

At line 123. Having the "(figure 1d)" at the end of the sentence makes it seem like this figure is transcriptomics which it is not. Consider moving the reference to the figure up to "ROS production in leaves" at line 120.

At line 209: "plants with the LeafSC" – it may help the reader to specify this is both genotypes of plant - consider changing to "Col-0 and rbohD plants with the LeafSC".

Parts of Figure 7 are a little too small – particularly parts a and c. In part c the open vs. filled circles for CAZyme vs. non-CAZyme seem incorrect: gspE is not a CAZyme and is an open circle? Maybe the filled circle is just not visible?

In Figure 1d. For the colonization what is the mock that is determining the significance denoted by the asterisk? Is it completely uninoculated plants? It seems unusual that there is an asterisk in every total colonization condition.

In Figure S6b. Which is the L148 alone and L148+LeafSC condition picture. Is it L148 P1 or L148 P9?

Reviewer #3 (Remarks to the Author):

Manuscript summary:

This manuscript aims to understand the molecular mechanisms underlying how plants exert immunity against potentially harmful pathogen while maintain commensal bacterial community. The authors focus to study the impact of RBOHD-mediated ROS in commensal bacteria isolated from healthy Arabidopsis plants and identify a specific commensal, Xanthomonas L148, which its in planta colonization was dramatically increased in Arabidopsis rbohD mutant and led to host mortality (disease symptom) in contrast to asymptomatic wild-type Col-0 plants. By Tn5 mutagenesis screen, the authors identify 18 bacterial mutants exhibited consistent loss-of-rbohD mortality. Further in-depth study on gspE, encoding a type II secretion system (T2SS) component, by characterizing the gspE::Tn5 and two gspE deletion mutants, the authors provided strong evidence that T2SS is required for the disease symptom of Xanthomonas L148 on rbohD mutant. Further in planta bacterial transcriptomics revealed that RBOHD suppresses expression of T2SS genes and L148 colonization protected plants against a bacterial pathogen Pseudomonas syringae pv, tomato (Pto) in T2SS-independent manner. Collectively, this study uncovered that RBOHD-mediated ROS can suppress the T2SS of a potentially harmful Xanthomonas L148, thereby converting it into a commensal that provides beneficial to host plants for increased resistance against the bacterial pathogen Pto.

Overall, this is a very comprehensive study which revealed a new mechanism explaining the host plant is able to use ROS to counter-defend the invasion of a potentially harmful pathogen

Xanthomonas L148 by dampening its T2SS expression while maintain commensal bacterial community. Previous observation by Pfeilmeier (2021) also identified Xanthomonas L131, a closely related strain of L148, can exert its detrimental impact on rbohD mutant plants. Thus, RbohD ROS-mediated counter-defense may be a common mechanism deployed by plants to tame a potentially detrimental leaf commensal and turns it into a microbe beneficial to the host. The experimental designs and methods are logical and sound, and the data are in high quality, which provide strong evidence to explain the results and conclusions drawn. I only have a few minor comments for the authors to consider for further improvement.

General comments:

While I enjoy reading the manuscript, I feel some statements in the “Results” may be more appropriate to be moved to “Discussion” to keep the results neutral without too much influence by the explanation led by authors. For examples, the statement “On the other hand, only heat-killed but not live Burkholderia L177 triggered ROS production, suggesting that L177 possesses MAMP(s) that are potentially recognized by plants but live L177 does not expose such MAMPs.” (Line 127-129) may delete the part “suggesting---”. The authors may discuss the potential mechanisms explaining why L177 does not expose such MAMPs and consider the alternative explanation such as that L177 may suppress PTI. Another example is the statement “These findings suggest that Arabidopsis recognizes commensal bacteria and produces ROS that does not have a detectable impact on most commensal bacterial colonization, at least in mono-associations.” (Line 160-161) should be moved to “Discussion”. I also don't quite agree with this explanation since Pto colonization was also not impacted by ROS (Kadota et al, 2014, also described in Introduction). So, why it is because of recognition?

Specific comments:

1. Although secretion of CAZyme is the most plausible cause of disease symptom of Xanthomonas L148 on rbohD mutant, it would be great if the authors can provide additional evidence for direct proof. For examples, can the loss of disease symptom of T2SS mutant on rbohD mutant be rescued by addition of these enzymes? Alternatively, if co-inoculation of gspE::Tn5 with either alaA::Tn5 or rpfF::Tn5 (T2SS is still function) can restore the disease symptom that is lost by inoculation of each mutant can also strengthen this argument.

2. Line 135-137 “This EFR dependency for commensal bacteria-induced ROS production was observed for other strains, but we detected no or only weak effects of mutations in FLS2 and CERK1.”. I am confused with this statement since the data seem opposite as most strains have strong response in fls2 and cerk1 mutants.

3. Line 294-295 “Both of the gspE deletion mutants as well as the gspE::Tn5 mutant showed loss of secretion activities and failed to cause disease in rbohD plants (Figure 6a-b).” does not seem to correlate with the data in Figure 6. The plate assay results seem to show some residual activities for peptin degradation by gspE::Tn5, but not so by two gspE deletion mutants? How do the authors quantify the activity and what is the explanation for the discrepancy between Tn5 and deletion mutant?

4. Line 365-368 “Further, Col-0 and rbohD plants pre-colonized with gspE::Tn5 had slightly better plant performance than the non-inoculated plants after Pto challenge (Supplementary Figure S10a-b), suggesting that the strain promotes plant fitness in the presence of pathogens.” is very interesting. In addition, the mutant seems to have better protection than L148 against endophytic Pto even though its colonization efficiency is lower than L148. I think this is important information that should be described in the results and discussed in the discussion. This brings up an issue that colonization efficiency is not always correlated with higher impacts for protection. Is it possible that the lack of T2SS can promote this bacterium with better plant fitness and then lead to higher protection ability than L148?

5. The authors may discuss the potential signal transduction pathway underlying how RBOHD-mediated ROS production suppresses expression of most T2SS gene, based on the current

knowledge.

6. Figure 5f: I would suggest to label each strain as 1, 2, ----8 besides their identity directly on the figure in order to correlate with the bacterial strains used in panel e. Please also indicate #9 as a mock control directly on the figure.

REVIEWER COMMENTS

Reviewer #1 (Remarks to the Author):

General assessment

The paper "Commensal lifestyle regulated by a negative feedback loop between Arabidopsis ROS and the bacterial T2SS." discusses the interaction between Arabidopsis and a Xanthomonas strain called L148. The study explores the relationship between reactive oxygen species (ROS) produced by Arabidopsis repress the bacterial type II secretion system (T2SS) of a commensal Xanthomonas isolate. The authors propose that there is a negative feedback loop between Arabidopsis ROS and the T2SS, which helps regulate the commensal lifestyle of the bacteria on plant leaves. The paper is clearly written and the topic interesting, however I do have some reservations that in my opinion should prevent publication in Nature communications:

Our response 1

We thank Reviewer 1 for the critical comments and for appreciating the clarity of writing and the substance of our research topic.

1) Novelty. Although the authors do cite Pfeilmeier et al (reference 17), I don't think they give sufficient credit to that paper. This paper already identified rbohD as a very important gene in retaining homeostasis in the leaf microbiota, and showed that especially Xanthomonas leaf148 and leaf131 were suppressed by RBOHD and become pathogens on rbohD mutant plants. I feel the authors of this study claims these discoveries too much as their own.

Our response 2

We think that we have given sufficient credit to Pfeilmeier et al, 2021 since their work was cited 7 times in the manuscript. We also cited their recent work which corroborates and complements our findings as well (Please see Our response 5). We think that these suffice.

2) The true novelty lies in the mechanism by which L148 kills the immunocompromised rbohD plant. Through a mutant screen the authors identify 214 candidate genes involved in pathogenicity and focus on the T2SS of L148. They show L148 growth is insensitive to ROS, but that T2SS is suppressed in WT and not in rbohD mutant plants. However a large amount of bacterial genes is differential expressed between the two plant genotypes (almost 75%) . This suggests that the ROS do have very brought spectrum effects and that L148 is not so insensitive to ROS.

Our response 3

We appreciate that Reviewer 1 finds novelty in our mechanistic rendition of the interaction of a potentially pathogenic commensal with the plant host. For details, please see Our response 7.

3) Related, unless there is a typ-O, I think the claim that "DEGs significantly enriched for the candidate genes detected from the Xanthomonas L148::Tn5 mutant screening" (line 318) is not true. 2946 DEGs out of a total of 4,208 genes (according to genbank) in the L148 is almost 75% of the genes of L148. That would mean that you would expect to

find more than 75 % of the 214 candidate genes from the mutant screen if there is over representation and 102 genes is a significant underrepresentation. I found in general the applied statistics were opportunistic and sloppy.

Our response 4

Please see Our response 17. We have revised the sentence according to the suggestion.

4) Moreover, T2SS are very common among gram negative bacteria and it is widely demonstrated that T2SS mediated secretion of extracellular enzymes is important for bacteria in natural environments. It seems to me that, even though the *gspE::tn5* (T2SS) mutant was not impaired in its growth in liquid growth medium, it might still very well be growth impaired in planta. I'm having a hard time imaging that the T2SS is what makes this *Xanthomonas* specifically pathogenic to *rbohD* mutant plant. My conclusion would thus be yes the T2SS is required for full virulence of L148 as L148 cannot grow properly without it, but this not likely the reason why L148 has this effect on *rbohD* mutant plant and I don't think this paper is (line 390) "is a significant advance in our understanding of how plant immune responses manipulate bacterial growth and behavior".

Our response 5

We respectfully disagree. We pointed out in the "Results" Figure 4b that the *gspE::Tn5* mutant has no colonization defects in Col-0 when compared to the wild-type L148 for both leaf compartments, only that *gspE::Tn5* mutant has a significantly decreased colonization in *rbohD* plants as compared to wildtype L148 at 3 dpi. This shows that the *gspE::Tn5* mutant does not have *in planta* growth aberrations in the context of the wild-type Col-0 plants. This observation is not merely due to persistence or variations in the initial inoculum load since bacterial populations at 3 dpi are significantly higher than 0 dpi, indicating active proliferation of both the wild-type and mutant strains (Line 270-278).

Pfeilmeier et al. 2023. bioRxiv had also independently identified T2SS genes in the *xps* operon as causal to the leaf degrading capacity of *Xanthomonas* L131 and L148 through targeted reverse genetics (in-frame deletion) since T2SS is a common apparatus for virulence deployed by many phytopathogenic *Xanthomonas* species (<https://doi.org/10.1101/2023.05.09.539948>, Figure 3) and were also captured in their L131 Tn5 library screening for leaf degradation (*xpsE* and *xpsD*), which further corroborates the results of our unbiased forward genetic screening of *Xanthomonas* L148 Tn5 mutant library and our targeted deletion for validation. As such, we have included the mentioned pre-print in our citations.

It now reads in the "Discussion" as (Line 442-443):

"Consistently, the same gene was recently found to be causal for leaf degradation and virulence of *Xanthomonas* L148 in *rbohD* plants, functioning to releasing substrates for other leaf commensals consequentially driving the phyllosphere microbiota composition⁵³."

There is limited literature about the effects of plant immunity on the bacteria. We believe that our paper is among the few papers in the plant field that have revealed the

molecular mechanism by which the plant host modulates the behavior of a potentially pathogenic commensal which is a bona-fide member of its native microbiota. With that we have revised our statement in the “Discussion” as follows (Line 391-392):

“This is a significant advance in our understanding of how plant immune responses manipulate the growth and behavior of its potentially pathogenic microbiota members.”

Detailed comments

Line 27-30: We found that commensal bacteria isolated from healthy *Arabidopsis* plants trigger diverse patterns of reactive oxygen species (ROS) production via the NADPH oxidase RBOHD that selectively inhibited specific commensals, notably *Xanthomonas* L148. - ->this is not something the authors found, it was previously found by: <https://doi.org/10.1038/s41564-021-00929-5>

Our response 6

They used only heat-killed cells of leaf isolates (Pfeilmeier et al, 2021. *Nature Microbiology*, <https://doi.org/10.1038/s41564-021-00929-5>, Figure 4c and Extended Data Figure5). We used both live and heat-killed bacterial cells to reveal potential immune-evasion or suppression signatures as we emphasized in the “Results” and touched in the “Discussion”. We also identified the immune-receptor dependencies of the ROS outburst for the strains tested by using a handful of PTI (co)receptor mutants (Please see Supplementary Figure S2). With this, we have now cited their work accordingly in “Results” and have revised the “Abstract” accordingly.

Now the “Abstract” reads as (Line 28-32):

“We found that commensal bacteria isolated from healthy *Arabidopsis* plants trigger diverse patterns of reactive oxygen species (ROS) production dependent on the immune receptors and completely on the NADPH oxidase RBOHD that selectively inhibited specific commensals, notably *Xanthomonas* L148.”

The “Results” reads as (Line 123-124):

“These commensal bacteria triggered diverse ROS production patterns which is consistent with the recent finding using heat-killed cells of leaf-isolated strains¹⁷.”

Line 34: In planta bacterial transcriptomics revealed that RBOHD suppresses most T2SS gene expression including *gspE*. - ->is this direct effect on T2SS or does it affect general metabolic activity?

Our response 7

The exact mechanism by which RBOHD-mediated ROS production leads to the suppression of T2SS genes is beyond the scope of the current study. Nevertheless, we attempted to address this through the pathway enrichment analysis in the “Results”

(Figure 7f, please refer to Supplementary Dataset S3 for the enriched GO terms) (Line 320-326).

Line 74 ROS also prohibits dysbiosis in *A. thaliana* leaves by suppressing *Xanthomonas*17- ->*Xanthomonas* is a bit simplistic. This paper showed that for the actual *Xanthomonas* strain that is also investigated in this study is kept in check, preventing dysbiosis. It appears the authors try to sell that an original research find..

Our response 8

We think that our description in the “Introduction” reflects their findings. In this study, we do not claim it but we focus on the mechanism by which the plant host controls a potentially pathogenic microbiota member through RBOHD-T2SS/CAZyme plant-*Xanthomonas* L148 interaction framework.

Line 94o (T2SS) has been shown to be necessary for the pathogenesis of many 95 phytopathogenic bacteria and mainly functions to secrete enzymes to degrade host 96 barriers and promote virulence20 - ->the function of the t2ss is more general

Our response 9

We have removed “mainly” to avoid confusion, according to the suggestion.

Line 161-165. It should not come as a surprise to the authors that RBOHD suppresses L148, this was shown before and reported in Nature microbiology

Our response 10

When we obtained this result, the Nature Microbiology paper had not been published yet. Nevertheless, we did not mention that this finding is surprising reflecting that the paper is already published and we cited the work accordingly.

Line 224 “ we developed a protocol” - ->This reads like a simple adaptation of the protocol developed by Pfeilmeier et al.

Our response 11

In Pfeilmeier et al. 2021. Nature Microbiology, they have used gnotobiotic system utilizing calcine clay in microboxes which has limited scalability (5 plants per microbox). In contrast to these methods, our Tn5 library screening used intact axenic 2-week-old *rbohD* plants grown on 96-well plates with 0.5x MS + 1.0% sucrose + 0.8% agar, mono-inoculated with the mutant strains and directly assayed for loss-of-*rbohD* mortality after 5 dpi. We developed and optimized this high-throughput screening protocol specifically intended for our purpose.

Line 233 only 124 had transposon insertions in genes with functional annotations. - ->this is surprising! Why is this? Did many genes in the genomes miss an annotation or did you enrich for

Our response 12

Only 124 genes have functional annotation, the rest of the candidate genes are hypothetical proteins or the insertions are located in non-coding regions (Supplementary Dataset S1). We pursued genes with annotated functions so that we could generate a testable hypothesis.

Figure 4a-b. it is a bit weird to test this with ANOVA, you would want to know whether the mutant is affected in its ability to colonize and cause disease in *rbohD* plants. A simple t-test between wildtype and *rbohD* would be better and much more sensitive than an ANOVA with so many comparisons.

Our response 13

We wanted to see the differences on the colonization behavior of the bacterial mutants and the wild-type (as exemplified in Our response 5), which means many comparisons (L148, *gspE::Tn5*, *alaA::Tn5*, and *rpfF::Tn5*). In this case, we think that ANOVA is appropriate.

Line 251 it looks to me that *gspE::tn5* is impaired in its colonization, but the ANOVA is not sensitive enough to capture this as a result of the many comparisons. In any case it is ridiculous to do the ANOVA over both time points

Our response 14

Similar to Our response 13, as we had multiple comparisons, we think that ANOVA is appropriate. Comparing bacterial populations at 0 and 3 dpi is necessary to provide evidence that these strains did not simply persist on the leaf but significantly increased their population over time.

Lines 301-304: *Xanthomonas* L148 pathogenicity is exerted in the absence of ROS through RBOHD while our *in vitro* results do not indicate general cellular toxicity of ROS. Thus, it can be assumed that RBOHD-mediated ROS production suppresses virulence of *Xanthomonas* L148. - ->is this true? 2,474 downregulated genes points towards something massive experienced by the *Xanthomonas* in wildtype that is missing in *rbohD*. Besides RBOHD mediates ROS production but likely many other things in the plant.

Our response 15

In this context, it is not a conclusion but an assumption, and we think that we can assume it. Moreover, 1) RBOHD affects the expression of *gspE* in plants, 2) ROS affects the expression of *gspE* *in vitro*, and 3) *gspE* mutants show compromised virulence. Based on these findings, we think that our assumption holds true. To address the concern that the effect of ROS on L148 virulence, we added a sentence in Discussion. It reads (Line 421-423):

“While the role of plant signaling activated by ROS cannot be excluded⁴⁷⁻⁵², our *in vitro* experiments support the notion that ROS produced by RBOHD modulates the behavior of the *Xanthomonas* L148.”

Lines 317 Strikingly, most T2SS apparatus genes including *gspE* were down-regulated in Col-0 as compared to *rbohD*. - ->This is not ‘striking’ considering more than half the genome is downregulated

Our response 16

We have deleted “strikingly” from the text.

Line 318-302 I either don't understand or have trouble believing that "DEGs were significantly enriched for the candidate genes detected from the *Xanthomonas* L148::Tn5 mutant screening." According to genbank *Xanthomonas* L148 has 4,208 genes). 102 genes overlapping between the 214 candidate genes (almost half of the candidate genes) from the mutant screen and 2946 DEGs (3/4 of the total genes) would be a significant UNDER representation of the candidate genes (<https://systems.crump.ucla.edu/hypergeometric/index.php>)

Our response 17

Thank you for the comments. We have removed the sentences.

Fig. 8 Like fig 4, also here all treatments are compared to all by ANOVA, which reduces the sensitivity of the statistical test a lot. In this way, the authors can claim that "Interestingly, *rbohD* mutant plants pre-colonized with *gspE*::Tn5 strain showed increased resistance against *Pto*, resembling *Xanthomonas* L148 pre-colonized Col-0 plants (Figure 8a, c)." However, there is a clear difference in number of *Pto* CFUs between wild-type and mutant leaf148 in fig 8a, that would have been significant if the authors would have chosen the statistical test less opportunistically.

Our response 18

Similar to previous comments, as we have multiple comparisons, we think that ANOVA is appropriate. Nevertheless, to more accurately describe our result, we have removed the part " , resembling....".

Line 353-373 these experiments were all performed on axenic plants. *Xanthomonas* leaf148 does not kill Col-0 plants but does colonize it. Hence, here we were looking here at the invasive success of *Pto* on its host with a competition and comparing to the success without it. Although I find these results intriguing I refrain from calling it a beneficial bacterium. In this artificial situation, my expectation is that any commensal bacterium would benefit the plant by competitive exclusion of a pathogen and suggest that the authors prove it can protect a plant growing in natural soil if they want to call it a plant-beneficial bacterium.

Our response 19

We have revised the statement accordingly and replaced "beneficial" with "protective" as to describe *Xanthomonas* L148 in this context. Our "Results" now reads as (Line 361-363):

"In summary, these results suggest that RBOHD-produced ROS turns the potentially harmful *Xanthomonas* L148 into a protective bacterium against aggressive pathogen colonization."

Also, we have indicated in the "Discussion" that we reserve the mechanistic underpinnings of the protective function of *Xanthomonas* L148 as future thrust (Line 474-475):

“The mechanisms by which the ROS-tamed *Xanthomonas* L148 confers pathogen protection warrants further investigation.”

Lines 392-394 “We have demonstrated that plant ROS licenses co-habitation with a potentially detrimental *Xanthomonas* L148 while it trains to guard against aggressive leaf pathogens.” In line with my previous comment I think this is a bridge to far. Yes ROS keep the commensal in check and prevents it from becoming a pathogen and yes a plant colonized by a commensal will be less easily invaded than an axenic plant, but ROS do not “train L148 to guard against”.

Our response 20

We have revised the statement accordingly. Now it reads as (Line 394-395):

“We have demonstrated that plant ROS allows co-habitation with a potentially pathogenic leaf microbiota member, *Xanthomonas* L148 by modulating its behavior.”

Reviewer #2 (Remarks to the Author):

In the paper by Entila et al. a mechanism of modulation of the pathogenic phenotype of an otherwise commensal microbe *Xanthomonas* L148 by ROS production via RBOHD in *Arabidopsis* is discovered. This work does a significant amount of screening of ROS production and colonization for 20 strains in combination with various MAMPs across various immune compromised *Arabidopsis* genotypes. This uncovers that strain L148 is significantly enriched in the *rbohD* genotype deficient in ROS production and causes a pathogenic phenotype not observed in the Col-0 plants where ROS production via RBOHD is intact. Then the authors identify the genetic factors determining this phenotype in L148. Through transposon mutagenesis they identify hundreds of L148 mutant strains with a loss of the pathogenic phenotype on *rbohD* plants. They identify three strains in particular with insertions in *gspE*, *alaA*, and *rpfF* and show their phenotypes (biofilm formation, motility, CAZyme production, etc.). The authors focus on the *gspE* mutant to show that there is a loss of CAZyme secretion but not a loss in colonization. Using transcriptomics in L148 they show that many of the *gsp* and CAZyme genes are downregulated in Col-0 vs the *rbohD* genotype suggesting these genes are responsive to ROS production in the Col-0 genotype. This is developed into a model where ROS production regulates the *gsp* system in L148 and this prevents it from becoming pathogenic on plants.

Our response 21

We thank Reviewer 2 for the critical comments and the appreciation of the amount of work done in this research.

Major comments:

I think much of the language to describe the interaction between the plant and L148 suggests the plant is very active in the interaction to prevent L148 from becoming a pathogen.

Our response 22

We appreciate the comments. We have revised the statements accordingly. Please see Our response 23-29.

at line 105 “converting *Xanthomonas* L148 into a commensal”. “Converting”

Our response 23

We have revised the statement according to the suggestion, it now reads as (Line 103-106):

“Using a bacterial random mutagenesis screen and *in planta* bacterial transcriptomics, we revealed that RBOHD-mediated ROS directly suppresses the T2SS of a potentially harmful *Xanthomonas* L148, which makes *Xanthomonas* L148 a commensal.”

line 339-340 “thus tightly regulated by immunocompetent wild-type Col-0 plants to prevent disease progression”. “tightly regulated by”

Our response 24

We have changed "tightly regulated by" to "regulated in".

line 392-393 "licenses" and "trains"

Our response 25

We have revised the statement accordingly. Now it reads as (Line 394-395):

"We have demonstrated that plant ROS allows co-habitation with a potentially pathogenic leaf microbiota member, *Xanthomonas* L148 by modulating its behavior."

line 394: "the plant host constrains... for its own benefits"

Our response 26

We have revised the statement according to the suggestion, it now reads as (Line 395-397):

"Our results show that the plant host constrains proliferation of this microbiota member by means of ROS as a molecular message to allow cooperation and coexistence".

line 414: "the plant host selectively constrains"

Our response 27

We would like to keep this statement.

line 462: "targeted by the host"

Our response 28

We have moved the statements and we hope that our description is appropriate in this context. It now reads as (Line 445-451):

"The T2SS is often utilized by plant pathogens to deliver CAZymes which degrade plant cell walls, allowing host invasion and promoting disease³⁰. For instance, the T2SS allows the root commensal *Dyella japonica* MF79 to efficiently colonize the host and is required for virulence of pathogenic *Dickeya dadantii*^{24,41}. These, together with our findings, suggest an important role of the T2SS in the establishment of microorganisms in host tissues, making it conceivable that it is targeted by the host to manipulate microbial behavior."

The default state in nature is that the wild type plant is going to produce ROS. It is possible that the microbe is evolved to not activate its pathogenic secretion of CAZymes in this situation so that it can survive mutualistically with the plant. Much of this language should likely be reconsidered to discuss the coevolution of this interaction and the mutualistic balance. I think the authors capture this best in lines 448-453. This description makes both the plant and bacteria less active in controlling the other.

Our response 29

We are pleased with the comments that Reviewer #2 appreciated our interpretation. We hope that our modifications based on the suggestions above are appropriate.

Figure 5 is a nice collection of data, but it seems less central to the main story. Could parts or all of this be moved to supplemental information?

Our response 30

We think that the wealth of data and the exhaustive characterization of the mutants in Figure 5 strengthens our claim that the *gspE::Tn5* mutant is the most suitable candidate to pursue. With this, we would like to keep it as is.

In Figure S8 and approximately line 282: How was it decided which strains are pathogens? In particular how do we know that the Xma strain from the human microbiome is not a plant pathogen?

Our response 31

As we are not sure whether *X. massiliensis* is pathogenic to plants, we have removed it from the figure and text.

Supplementary figure 10 and the section starting at line 353. The interpretation of this data seems to go beyond what it shows. What happens if you pre-colonize with other commensals and then add Pto? Is this exclusion of Pto exclusive to L148 or would any other commensals fill this niche and exclude Pto? The description that L148 “promotes plant fitness in the presence of pathogens” (line 367) and that ROS turns L148 “into a beneficial bacterium, thereby protecting the plant from aggressive pathogen colonization” (line 372) seem to be giving L148 too much credit when no other strains were tested in this assay.

Our response 32

We have deleted “, suggesting that the strain promotes...” and modified the text. Now it read (Line 3361-363):

“In summary, these results suggest that RBOHD-produced ROS turns the potentially harmful *Xanthomonas* L148 into a protective bacterium against aggressive pathogen colonization.”

In the model of Figure 8c. I don't understand the microbiota suppressing L148 at the top center of the model. In figure S6 L148 always has an effect on *rbohD* plants with and without the SynCom. And in Col-0 plants there is no difference between L148 alone and in the SynCom.

Our response 33

As Col-0 plants inoculated with L148 performed slightly better in the presence of SynCom, we think that leaf commensals also weakly contributed to the suppression of L148 in Col-0 plants. To reflect this, we have introduced a sentence in the Figure 8 legends “The leaf microbiota weakly affects *Xanthomonas* L148 *in planta* (dashed line).”

In Figure S9. How are the “potential substrates” of these CAZymes determined? Some GH families have very broad substrate activity. For example GH5 can be active on beta-glucan, beta-mannan, xylan, etc. and is only listed as beta-mannan in the Figure S9a. It may be better to keep this entire figure in the context of the CAZyme family number annotation (GH, PL, CBM) and not suggest the substrates if they are not experimentally validated.

Our response 34

Thank you very much for the comment. We have now used the higher order categories for the potential substrates of CAZymes to reflect broad substrate activity of some GH5s. We have revised the Supplementary Figure S9 and S8 accordingly, in which we have used the refined categories for the substrates. We hope that they are appropriate now.

A few points in the methods are missing some critical detail to replicate the assays:
 Line 545: The description of how the flood inoculation is performed is not clear. How much volume of the OD=0.005 bacteria solution? How long is it placed in the plate?

Our response 35

We have revised the “Materials and Methods” accordingly and added specific details. It now appears as (Line 557-560):

“Two-week-old seedlings grown on 0.5x MS medium agar in a chamber at 23 °C/23 °C (day/night) with 10 h of light were flood-inoculated with these bacterial suspensions (15 mL per plate, incubated for 1 min), drained, air-dried for 5 min and then incubated in the same growth chamber.”

Line 588: 96 well plate assay. How much agar in each well? What are the plant growth conditions?

Our response 36

We have revised the “Materials and Methods” accordingly and added specific details. It now appears as (Line 602-604):

“Seedlings of *rbohD* were grown in 96-well plates with 500 μ L 0.5x MS agar with 1% sucrose for each well and incubated in a growth chamber at 23 °C/23 °C (day/night) with 10 h of light for 14 days.”

Line 648: How are the bacteria separated from the plant tissue specifically. What centrifugation speed and time?

Our response 37

We have revised the “Materials and Methods” accordingly and added specific details. It now appears as (Line 663-664):

“Bacterial cells were separated from the plant tissue through 6 μ m filter mesh and centrifugation at 1300g for 10s”.

Line 695: what “chemiluminescence” was used?

Our response 38

We apologize our mistake. The pathogen *Pto lux* was engineered to emit luminescence (Matsumoto et al Plant Commun 2022). We have revised the “Materials and Methods” to bring some clarity, it now reads as (Line 711-712):

“Colonies of the *Pto lux*⁶⁷ were differentiated against *Xanthomonas* L148 strains via their color and luminescence, and colonization was expressed as cfu mg⁻¹ leaf sample.”

Minor comments:

At line 123. Having the “(figure 1d)” at the end of the sentence makes it seem like this figure is transcriptomics which it is not. Consider moving the reference to the figure up to “ROS production in leaves” at line 120.

Our response 39

Done. Thank you.

At line 209: “plants with the LeafSC” – it may help the reader to specify this is both genotypes of plant - consider changing to “Col-0 and *rbohD* plants with the LeafSC”.

Our response 40

Done. Thank you.

Parts of Figure 7 are a little too small – particularly parts a and c. In part c the open vs. filled circles for CAZyme vs. non-CAZyme seem incorrect: *gspE* is not a CAZyme and is

an open circle? Maybe the filled circle is just not visible?
 Our response 41
 We have revised the Figure 7 to avoid confusion.

Figure 7

In Figure 1d. For the colonization what is the mock that is determining the significance denoted by the asterisk? Is it completely uninoculated plants? It seems unusual that there is an asterisk in every total colonization condition.

Our response 42

The asterisks denote that the bacterial colonization is significantly different from uninoculated plants (0 cfu/mg leaf) for the total colonization. It means that the commensals can colonize the total leaf compartment significantly. For the endophytic compartments, though we detected some of the commensals through plating techniques, the population was too small which makes differences from uninoculated

plants (0 colony) not significant. The disparities in the colonization patterns of the commensal give us insight on the preferential lifestyle of the strains: epiphytic and/or endophytic. We would like to keep the Figure 1d as is and we hope this is appropriate.

In Figure S6b. Which is the L148 alone and L148+LeafSC condition picture. Is it L148 P1 or L148 P9?

Our response 43

We have revised Supplementary Figure S6 accordingly for clarity.

Reviewer #3 (Remarks to the Author):

Manuscript summary:

This manuscript aims to understand the molecular mechanisms underlying how plants exert immunity against potentially harmful pathogen while maintain commensal bacterial community. The authors focus to study the impact of RBOHD-mediated ROS in commensal bacteria isolated from healthy Arabidopsis plants and identify a specific commensal, Xanthomonas L148, which its in planta colonization was dramatically increased in Arabidopsis rbohD mutant and led to host mortality (disease symptom) in contrast to asymptomatic wild-type Col-0 plants. By Tn5 mutagenesis screen, the authors identify 18 bacterial mutants exhibited consistent loss-of-rbohD mortality. Further in-depth study on gspE, encoding a type II secretion system (T2SS) component, by characterizing the gspE::Tn5 and two gspE deletion mutants, the authors provided strong evidence that T2SS is required for the disease symptom of Xanthomonas L148 on rbohD mutant. Further in planta bacterial transcriptomics revealed that RBOHD suppresses expression of T2SS genes and L148 colonization protected plants against a bacterial pathogen Pseudomonas syringae pv, tomato (Pto) in T2SS-independent manner. Collectively, this study uncovered that RBOHD-mediated ROS can suppress the T2SS of a potentially harmful Xanthomonas L148, thereby converting it into a commensal that provides beneficial to host plants for increased resistance against the bacterial pathogen Pto.

Overall, this is a very comprehensive study which revealed a new mechanism explaining the host plant is able to use ROS to counter-defend the invasion of a potentially harmful pathogen Xanthomonas L148 by dampening its T2SS expression while maintain commensal bacterial community. Previous observation by Pfeilmeier (2021) also identified Xanthomonas L131, a closely related strain of L148, can exert its detrimental impact on rbohD mutant plants. Thus, RbohD ROS-mediated counter-defense may be a common mechanism deployed by plants to tame a potentially detrimental leaf commensal and turns it into a microbe beneficial to the host. The experimental designs and methods are logical and sound, and the data are in high quality, which provide strong evidence to explain the results and conclusions drawn. I only have a few minor comments for the authors to consider for further improvement.

Our response 44

We thank Reviewer 3 for the critical comments and for appreciating the quality and rigor of our research.

General comments:

While I enjoy reading the manuscript, I feel some statements in the “Results” may be more appropriate to be moved to “Discussion” to keep the results neutral without too much influence by the explanation led by authors. For examples, the statement “On the other hand, only heat-killed but not live Burkholderia L177 triggered ROS production, suggesting that L177 possesses MAMP(s) that are potentially recognized by plants but live L177 does not expose such MAMPs.” (Line 127-129) may delete the part “suggesting---“. The authors may discuss the potential mechanisms explaining why

L177 does not expose such MAMPs and consider the alternative explanation such as that L177 may suppress PTI.

Our response 45

We have removed “, suggesting that...” as suggested. We decided not to include a discussion on this as it is too speculative.

Another example is the statement “These findings suggest that *Arabidopsis* recognizes commensal bacteria and produces ROS that does not have a detectable impact on most commensal bacterial colonization, at least in mono-associations.” (Line 160-161) should be moved to “Discussion”.

Our response 46

We have moved the statement to Discussion (Line 374-376), according to the suggestion.

I also don't quite agree with this explanation since Pto colonization was also not impacted by ROS (Kadota et al, 2014, also described in Introduction). So, why it is because of recognition?

Our response 47

To explain this better, we have elaborated our “Discussion” in this regard, now it reads as (Line 374-389):

“These findings suggest that *Arabidopsis* recognizes commensal bacteria and produces ROS that does not have a detectable impact on most commensal bacterial colonization. This highlights the notion that the perception of the microbial signal, followed by the cascade of immune signals, and immune execution leading to the restriction of microbial colonization are distinct events. This corroborates our previous finding that plant and transcriptome responses of commensal bacteria are largely uncoupled during an early stage of infection²⁶. Plant responses to MAMPs do not necessarily affect the consequence of plant-microbe interactions. This makes sense because MAMPs derived from pathogenic or mutualistic bacterium can induce the same PTI responses³⁴⁻³⁵. Also, different types of MAMPs or its variants can elicit quantitatively different intensities of immune responses and detection of these molecular patterns might not essentially result in the same immune outputs and different immune readouts are activated by different MAMPs or to the catalogue of MAMPs that a particular microbe possesses³⁶⁻³⁹. Thus, the perception of the nuanced compendium of stimuli, triggering the cascade of signals, and the eventual impact of plant immune responses on microbes are tailored for each microbial strain.”

Specific comments:

1. Although secretion of CAZyme is the most plausible cause of disease symptom of *Xanthomonas* L148 on *rbohD* mutant, it would be great if the authors can provide additional evidence for direct proof. For examples, can the loss of disease symptom of T2SS mutant on *rbohD* mutant be rescued by addition of these enzymes? Alternatively, if co-inoculation of *gspE::Tn5* with either *alaA::Tn5* or *rpfF::Tn5* (T2SS is still function)

can restore the disease symptom that is lost by inoculation of each mutant can also strengthen this argument.

Our response 48

We consider that the identification of the actual mechanism of L148 virulence on *rbohD* plants is beyond the scope of the current study. As *Xanthomonas* L148 potentially encodes over 100 CAZymes, it will be a long way to identify which one is the cause. In addition, there can be multiple CAZymes responsible for L148 virulence. While *alaA::Tn5* and *rpfF::Tn5* secreted CAZymes *in vitro*, they may not do so in plants. In addition, to complement *gspE::Tn5* by *alaA::Tn5* and *rpfF::Tn5*, they would need co-colonize closely. In our low dose of inoculation system, this unlikely occurs. Finally, even *alaA::Tn5* and *rpfF::Tn5* complement *gspE::Tn5*, it does not directly demonstrate that CAZymes are the cause for L148 virulence. Due to these reasons, we decided not to perform the suggested experiments but will look into this in the future.

2. Line 135-137 “This EFR dependency for commensal bacteria-induced ROS production was observed for other strains, but we detected no or only weak effects of mutations in FLS2 and CERK1.”. I am confused with this statement since the data seem opposite as most strains have strong response in *fls2* and *cerk1* mutants.

Our response 49

We have revised the statement according to the suggestion and now reads as (Line 134-137):

“This EFR dependency for commensal bacteria-induced ROS production was observed for other strains, since many of which had reduced or lost ROS burst for *efr* leaf disks but considerable ROS signals were detected for *fls2* and *cerk1* (Supplementary Figure S2).”

3. Line 294-295 “Both of the *gspE* deletion mutants as well as the *gspE::Tn5* mutant showed loss of secretion activities and failed to cause disease in *rbohD* plants (Figure 6a-b).” does not seem to correlate with the data in Figure 6. The plate assay results seem to show some residual activities for peptin degradation by *gspE::Tn5*, but not so by two *gspE* deletion mutants? How do the authors quantify the activity and what is the explanation for the discrepancy between Tn5 and deletion mutant?

Our response 50

The enzymatic indices were calculated by dividing the zones of clearing by the colony size. Consistently, both the *gspE::Tn5* and the *gspE* deletion mutants lost the capacity to degrade milk, however we observed slight disparities when it comes to pectin. We think that this is attributable to background activity or trait variability. Nevertheless, all of the *gspE* mutant strains had significantly reduced pectin degradation as compared to the *Xanthomonas* L148 wild-type. To more accurately describe our result, we have modified the text accordingly. Now it reads (Line 292-294):

“Both of the *gspE* deletion and the *gspE::Tn5* mutants showed reduced secretion activities and failed to cause disease in *rbohD* plants (Figure 6a-b).”

We have also included a statement reflecting the bioRxiv paper by Pfeilmeier et al, 2023 published on the same day with ours (<https://doi.org/10.1101/2023.05.09.539948>, Figure 3) in Discussion. It reads (Line 442-445):

“Consistently, the same gene was recently found to be causal for leaf degradation and virulence of *Xanthomonas* L148 in *rbohD* plants, functioning to help releasing substrates for other leaf commensals, consequentially driving the phyllosphere microbiota composition⁵³.”

4. Line 365-368 “Further, Col-0 and *rbohD* plants pre-colonized with *gspE::Tn5* had slightly better plant performance than the non-inoculated plants after Pto challenge (Supplementary Figure S10a-b), suggesting that the strain promotes plant fitness in the presence of pathogens.” is very interesting. In addition, the mutant seems to have better protection than L148 against endophytic Pto even though its colonization efficiency is lower than L148. I think this is important information that should be described in the results and discussed in the discussion. This brings up an issue that colonization efficiency is not always correlated with higher impacts for protection. Is it possible that the lack of T2SS can promote this bacterium with better plant fitness and then lead to higher protection ability than L148?

Our response 51

We have touched this in our “Discussion”, and now appears as (Line 469-474):

“The *gspE::Tn5* mutant provided better pathogen protection to Col-0 plants than the wild-type *Xanthomonas* L148 despite its lesser leaf colonization, suggesting that efficient colonization does not necessarily translate to pathogen protection. This finding is in line with the recent finding in which a number of microbiota strains that robustly colonize did not confer plant protection against pathogen attack³³.”

5. The authors may discuss the potential signal transduction pathway underlying how RBOHD-mediated ROS production suppresses expression of most T2SS gene, based on the current knowledge.

Our response 52

We thank Reviewer 3 for pointing this out. We have attempted to address the potential signaling cascades underlying the suppression of bacterial T2SS expression via RBOHD-mediated plant ROS production from our *in planta* bacterial transcriptome, for which we have looked into bacterial transcription factors involved in oxidative stress, in particular the OxyR, the H₂O₂ sensor transcription factor; SoxR, the O₃²⁻ sensor transcription factor, Crp, cAMP-RECEPTOR PROTEIN transcription factor, and Fur, FUMARATE-NITRATE REDUCTASE REGULATION transcription factor. We have also looked into the motif enrichment of these transcription factors on the promoters of *gspE*, *alaA*, *rpfF*, and a handful of other genes (representatives from the T2SS, and oxidative-stress responsive genes such as catalase *katG*, please see figure below). Though we find some patterns interesting, we think that it is too speculative to include in the result.

6. Figure 5f: I would suggest to label each strain as 1, 2, ---8 besides their identity directly on the figure in order to correlate with the bacterial strains used in panel e. Please also indicate #9 as a mock control directly on the figure.

Our response 53

Done. Thank you. Please see the new Figure 5 below.

REVIEWER COMMENTS

Reviewer #1 (Remarks to the Author):

I thank the authors for the efforts in improving the manuscript. They have satisfactorily addressed some of the points and this improved the manuscript. However, there are three issues with this manuscript that I previously raised, that the authors don't agree with and chose to ignore. My view point has not really changed, I grouped the author's responses per issue and tried to make myself more clear.

Issue 1) I think that in the introduction you should make explicit that the role of RBOHD in suppressing the growth and pathogenicity of your commensal *Xanthomonas* strain (L148) was already discovered. Line 74-75 [ROS also prohibits dysbiosis in *A. thaliana* leaves by suppressing *Xanthomonas*17.] is misleading in this as it does not mention RBOHD and L148. As a result, the manuscript now reads as if you started out with an uninformed screen of the effects of random set of bacteria on a selected set immunity impaired mutants and you pick up on L148 and RBOHD. The first paragraph of the results section is a validation of the earlier findings by Pfeilmeier, why not mention it? The fact that you did not know when you started these experiments (your response 10) is irrelevant in my opinion. This paper should make clear what is already known from literature and what it is that your study adds to that.

Issue 2) Indeed T2SS is a common apparatus that affects the virulence of phytopathogenic *Xanthomonas* species (response 5) but it is not specific to phytopathogens. It is widely conserved among gram negative bacteria and is unlikely only required for pathogenicity. The T2SS mediates extracellular delivery of a variety of toxins, lipases and enzymes that break down complex carbohydrates thus conferring a survival advantage to pathogenic as well as other environmental species. It is unlikely that T2SS defective mutants are not impaired in growth in a complex environment and your figure 4b actually show that the *gspE::tn5* is impaired in its colonization compared to L148 wildtype, also on wild-type plants. You just used an ANOVA that could never determine that.

Yes, an ANOVA is used for multiple comparisons (Response 13 and 14), but an ANOVA in which 40 treatments are compared (all bacterial genotypes, plant genotypes, time points and leaf compartments are compared with each other) is very insensitive. It is wrong to conclude that *gspE::Tn5* mutant does not have in planta growth aberrations in the context of the wild-type Col-0 plants based on this test.

The authors claim that it is necessary to compare "bacterial populations at 0 and 3 dpi [...] to provide evidence that these strains did not simply persist on the leaf but significantly increased their population over time" (response 14). I don't think that is really important here. Clearly the bacteria grew from 0dpi to 3dpi, they increased orders of magnitude (1000-10000 times), you don't need the statistics to draw that conclusion. It is more important that you correctly assess the difference between the L148 WT and mutants. Moreover including multiple timepoints ruins not only the sensitivity of the ANOVA, but also ensures that the assumption of equal variance across populations is definitely not met (there is no variance in most samples at 0 dpi). In this way the ANOVA cannot be used. The ANOVA in figure 4b should be at least limited to compare only the abundance of L148 and its three mutant derivatives on 1 plant genotype and on 1 time point at a time. If then there is no significant difference, between L148 and *gspE::Tn5* mutant, I would be more convinced that there are no growth aberrations, even though 3-4 replicates is a real low number to demonstrate a difference in growth.

Issue 3) The authors use in planta bacterial transcriptomics to confirm the result of their

mutant screen. In the original paper, the authors erroneously claimed that the 214 genes identified by screening the L148 mutant library were overrepresented amongst the differentially expressed genes (DEGs) identified in their transcriptomics analysis. I pointed out that it is the opposite, the 214 genes are significant underrepresented among the DEGs. In response 4 the authors say. We have revised the sentence according to the suggestion. However, they have removed the analysis completely from the text. I think this analysis is nonetheless important as it shows that the mutant screen is not in line with the transcriptomics data, and this should not be ignored because it is inconvenient for the story line.

In addition, the authors focus on that their finding that the T2SS genes are suppressed in WT and not in *rbohD* mutant plants. However, as mentioned, a large amount of bacterial genes is differential expressed between the two plant genotypes (almost 75% of the total number of genes). This suggests that the ROS do have very brought spectrum effects and that L148 is not so insensitive to ROS. Yes, "most T2SS apparatus genes including *gspE* were down-regulated in Col-0 as compared to *rbohD* (Figure 7c–e)" (line 315), but as the large majority of L148 genes is suppressed this would be true for almost 75% of the gene clusters.

All in all, I am not convinced by the data represented here that the T2SS system is the defining property of L148 that makes it damaging *rbohD* mutant plants, nor do I think that the T2SS system is specifically affected by the ROS in wildtype plants. Hence I don't think there is a a negative feedback loop between Arabidopsis ROS and the bacterial T2SS.

Reviewer #2 (Remarks to the Author):

The authors have addressed all of my comments.

A few minor edits:

At line 183 "ROS concentrations up to 1mM"

1mM of what? H₂O₂? paraquat? I find this confusing.

Line 233. Is it possible to direct the reader to which strains are the 18 strains? what genes are disrupted in them? I think this would be helpful. Is this in a supplementary table?

The labeling on the circle in Figure 3b is overlapping and not possible to read in places. This should be fixed.

Reviewer #3 (Remarks to the Author):

The authors have substantially addressed my concerns. Just one minor point. Please fix the problems of overlapping text in Fig. 3b by labelling the gene names properly.

REVIEWER COMMENTS

Reviewer #1 (Remarks to the Author):

I thank the authors for the efforts in improving the manuscript. They have satisfactorily addressed some of the points and this improved the manuscript. However, there are three issues with this manuscript that I previously raised, that the authors don't agree with and chose to ignore. My view point has not really changed, I grouped the author's responses per issue and tried to make myself more clear.

Our response 1

We appreciate Reviewer #1 comments to greatly improve our manuscript.

Issue 1) I think that in the introduction you should make explicit that the role of RBOHD in suppressing the growth and pathogenicity of your commensal *Xanthomonas* strain (L148) was already discovered. Line 74-75 [ROS also prohibits dysbiosis in *A. thaliana* leaves by suppressing *Xanthomonas*¹⁷.] is misleading in this as it does not mention RBOHD and L148. As a result, the manuscript now reads as if you started out with an uninformed screen of the effects of random set of bacteria on a selected set immunity impaired mutants and you pick up on L148 and RBOHD. The first paragraph of the results section is a validation of the earlier findings by Pfeilmeier, why not mention it? The fact that you did not know when you started these experiments (your response 10) is irrelevant in my opinion. This paper should make clear what is already known from literature and what it is that your study adds to that.

Our response 2

We have revised the Introduction according to the suggestion. It now reads as (Line74-77):

“RBOHD also prohibits dysbiosis in *A. thaliana* leaves by suppressing *Xanthomonas* L131 and the close strain L148, as indicated by their domination in phyllosphere microbiota associated with disease onset on *rbohD* mutant plants upon inoculation with synthetic bacterial communities¹⁷.”

Regarding the difference in ROS burst experiments between this study and Pfeilmeier *et al.*, we tested both living and heat-killed dead bacteria while Pfeilmeier *et al.* used only heat-killed dead bacteria. In addition, we included MAMP receptor mutants for ROS burst assays in contrast to Pfeilmeier *et al.* Thus, our analysis is not simply a repeat of their experiments. We hope that it is clear in the text (Line 116-127).

Issue 2) Indeed T2SS is a common apparatus that affects the virulence of phytopathogenic *Xanthomonas* species (response 5) but it is not specific to phytopathogens. It is widely conserved among gram negative bacteria and is unlikely only required for pathogenicity. The T2SS mediates extracellular delivery of a variety of toxins, lipases and enzymes that break down complex carbohydrates thus conferring a survival advantage to pathogenic as well as other environmental species. It is unlikely that T2SS defectives mutants are not impaired in growth in a complex environment and your figure 4b actually show that the *gspE::tn5* is impaired in its colonization compared

to L148 wildtype, also on wild-type plants. You just used an ANOVA that could never determine that.

Yes, an ANOVA is used for multiple comparisons (Response 13 and 14), but an ANOVA in which 40 treatments are compared (all bacterial genotypes, plant genotypes, time points and leaf compartments are compared with each other) is very insensitive. It is wrong to conclude that *gspE::Tn5* mutant does not have in planta growth aberrations in the context of the wild-type Col-0 plants based on this test.

The authors claim that it is necessary to compare “bacterial populations at 0 and 3 dpi [...] to provide evidence that these strains did not simply persist on the leaf but significantly increased their population over time” (response 14). I don't think that is really important here. Clearly the bacteria grew from 0dpi to 3dpi, they increased orders of magnitude (1000-10000 times), you don't need the statistics to draw that conclusion. It is more important that you correctly assess the difference between the L148 WT and mutants. Moreover including multiple timepoints ruins not only the sensitivity of the ANOVA, but also ensures that the assumption of equal variance across populations is definitely not met (there is no variance in most samples at 0 dpi). In this way the ANOVA cannot be used. The ANOVA in figure 4b should be at least limited to compare only the abundance of L148 and its three mutant derivatives on 1 plant genotype and on 1 time point at a time. If then there is no significant difference, between L148 and *gspE::Tn5* mutant, I would be more convinced that there are no growth aberrations, even though 3-4 replicates is a real low number to demonstrated a difference in growth.

Our response 3

According to the suggestion, we have done ANOVA within genotype, compartment, and day-post-inoculation as suggested. We have found that the *gspE::Tn5* mutant have a lower Col-0 total leaf colonization as compared to the wild-type L148 ($P < 0.05$). However, Col-0 endophytic colonization for both *gspE::Tn5* mutant and wild-type L148 are not statistically different at 3 dpi ($P > 0.05$). Accordingly, we have modified Figure 4b.

Our conclusions were not made based on no statistical difference in Col-0 total colonization between *gspE::Tn5* mutant and wild-type L148. Our conclusions were based on: 1) wild type L148 colonized more on *rbohD* compared with Col-0 plants and causes disease on *rbohD* but not Col-0 plants 2) *gspE::Tn5* mutant colonized in *rbohD* compared with Col-0 plants less than wild type L148 and did not cause disease on *rbohD* plants. 3) T2SS was compromised in the *gspE* mutant 4) RBOHD-dependent ROS production triggered by living *gspE::Tn5* mutant was less than that by wild-type L148. 5) RBOHD inhibited the expression of T2SS including *gspE in planta* and ROS inhibits the expression of *gspE in vitro*. Thus, our conclusions remain unchanged.

The Results now read as (Line251-254).

“By contrast, *gspE::Tn5* mutant exhibited comparable endophytic colonization and slightly lower total leaf colonization capacities to *Xanthomonas* L148 wild-type in Col-0 leaves, but failed to show increased colonization in *rbohD* compared with Col-0 plants in contrast to wild-type L148 (Figure 4b).”

(Line258-260)

“This indicates that *gspE::Tn5* mutant retains its endophytic colonization ability, while its capacity to more efficiently colonize *rbohD* plants is compromised compared to wild-type L148.”

We have modified the Discussion (Line470-475).

“The *gspE::Tn5* mutant had similar leaf endophytic but slightly reduced total colonization in wild-type Col-0 plants compared with wild-type L148. However, the *gspE::Tn5* mutant did not show increased colonization in *rbohD* in contrast to wild-type L148 (Figure 4b). This suggests that ROS functions to attenuate T2SS activity albeit incomplete and makes *Xanthomonas* L148 a commensal bacterium in wild-type Col-0 plants.”

Issue 3) The authors use in planta bacterial transcriptomics to confirm the result of their mutant screen. In the original paper, the authors erroneously claimed that the 214 genes identified by screening the L148 mutant library were overrepresented amongst the differentially expressed genes (DEGs) identified in their transcriptomics analysis. I pointed out that it is the opposite, the 214 genes are significant underrepresented among the DEGs. In response 4 the authors say. We have revised the sentence according to the suggestion. However, they have removed the analysis completely from the text. I think this analysis is nonetheless important as it shows that the mutant screen is not in line with the transcriptomics data, and this should not be ignored because it is inconvenient for the story line.

Our response 4

According to the suggestion, we have included the enrichment analysis and revised the Results as follows (Line321-324).

“Differentially expressed genes in Col-0 compared with *rbohD* were captured as candidate genes in the Tn5 mutant screening (73 down-regulated and 29 up-regulated out of 214 candidates), but these candidate genes were rather under-represented (hypergeometric test, $p\text{-value}=1.00\text{E-}10^{*}$).”**

In addition, the authors focus on that their finding that the T2SS genes are suppressed in WT and not in *rbohD* mutant plants. However, as mentioned, a large amount of bacterial genes is differential expressed between the two plant genotypes (almost 75% of the total number of genes). This suggests that the ROS do have very brought spectrum effects and that L148 is not so insensitive to ROS. Yes, “most T2SS apparatus genes including *gspE* were down-regulated in Col-0 as compared to *rbohD* (Figure 7c–e)”(line 315), but as the large majority of L148 genes is suppressed this would be true for almost 75% of the gene clusters.

Our response 5

Massive transcriptome change might suggest the versatile roles of RBOHD in regulating bacterial responses in plants. This suggests that suppression of T2SS expression is one of the causes that distinguish L148 colonization in Col-0 and *rbohD* plants but not the only one. Indeed, we identified more L148 mutants that failed to show increased colonization and cause disease in *rbohD* plants. Nevertheless, the fact that expression

of T2SS is suppressed by *RBOHD* in plants and ROS *in vitro* and *gspE* mutants showed failed to show increased colonization and cause disease in *rbohD* plants suggests that suppression of T2SS is one of the causes. This is analogous to the observation of that massive transcriptional changes occur in plants upon pathogen infection (over 10000 differentially regulated genes) and that a single mutation in a gene can cause an immune phenotype.

We have indicated these points more clearly in the Results (Line315-321).

“Statistical analysis revealed 2,946 differentially expressed genes (DEGs) upon comparing *in planta* bacterial transcriptomes in Col-0 with *rbohD* leaves (threshold: q-values < 0.05): 563 genes were up-regulated and 2,474 genes were down-regulated in Col-0 compared with *rbohD* plants, indicating global bacterial transcriptomic re-configuration upon leaf colonization as influenced by RBOHD (Figure 7a and c, See Supplementary Dataset S2 for the details on DEGs).”

and in the Discussion (Line430-434)

“While the massive bacterial transcriptomic landscape reprogramming can be attributed to versatile roles of RBOHD⁴⁷ such as cell wall remodelling⁴⁸⁻⁴⁹, host signalling⁵⁰, and stomatal response⁵¹⁻⁵², our *in vitro* experiments support the notion that ROS produced by RBOHD modulates the behavior of the *Xanthomonas* L148.”

We have also revised the subheading regarding *in vitro* ROS exposures, to clearly indicate that ROS does not significantly impact *Xanthomonas* L148 viability or growth *in vitro*, instead of “insensitive” (Line180).

“*Xanthomonas* L148 *in vitro* growth is largely unaffected by ROS”

All in all, I am not convinced by the data represented here that the T2SS system is the defining property of L148 that makes it damaging *rbohD* mutant plants, nor do I think that the T2SS system is specifically affected by the ROS in wildtype plants. Hence I don't think there is a a negative feedback loop between Arabidopsis ROS and the bacterial T2SS.

Our response 6

We think that the following results suggest that there is a negative feedback loop as the plant host controls the behavior of the potentially pathogenic leaf microbiota member through RBOHD-mediated ROS production. Thus, our major conclusions remain unchanged.

1. The *gspE::Tn5* mutant had lost its lethality towards *rbohD* plants.
2. The *gspE::Tn5* mutant triggered less ROS burst in live context but elicited similar ROS burst to the wild-type *Xanthomonas* L148 when MAMPs are heat-released in wild-type Col-0 plants.
3. The *gspE* gene expression is suppressed by RBOHD in plants and ROS *in vitro*.

In addition, an independent study from a different lab by Pfeilmeier et al, 2023. bioRxiv (<https://doi.org/10.1101/2023.05.09.539948>, Figure 3) had captured the same gene

(involved in T2SS) in their genetic screenings, which strongly corroborates and complements our findings as we described in the Discussion (Line453-456).

As mentioned, we are not claiming that T2SS is the sole defining bacterial factor for RBOHD-dependent pathogenicity, as the reviewer pointed out that there is a global bacterial transcriptome difference in Col-0 and *rbohD* plants and the fact that we identified more mutants than T2SS in our Tn5 library screening. This suggests multiple mechanisms for RBOHD-dependent virulence and T2SS is one of the causes. We have included the following statement in the Discussion (Line478-483).

“Our genetic screening and *in planta* bacterial transcriptome analysis (Figures 3b and 7a; Supplementary Dataset S1 and S2) also suggest that the conditional pathogenicity of *Xanthomonas* L148 depends on multiple pathways including amino acid metabolism (*alaA*, *DAO*, *speD*, and *prnA*), quorum sensing and bacterial signaling (*rpfF*, *rsbu_P*), surface adhesion (*sphB1*, *pilD*), and antioxidant metabolism (*ubiB*).”

Similar to any classical genetic studies, we think that when a gene mutation results in a loss of phenotype, the gene is then inferred causal. We think that our approach is not so different from other genetic studies. As mentioned, we think that we have provided sufficient evidence for a negative feedback loop between ROS and T2SS.

Reviewer #2 (Remarks to the Author):

The authors have addressed all of my comments.

Our response 7

We appreciate Reviewer #2 comments to greatly improve our manuscript.

A few minor edits:

At line 183 "ROS concentrations up to 1mM"

1mM of what? H₂O₂? paraquat? I find this confusing.

Our response 8

We have clarified the statement in the Result (Line185-187).

“To our surprise, *Xanthomonas* L148 seemed to tolerate acute treatments with ROS and retained viability up to H₂O₂ and O₂⁻¹ concentrations of 1 mM (Supplementary Figure S5a-b).”

Line 233. Is it possible to direct the reader to which strains are the 18 strains? what genes are disrupted in them? I think this would be helpful. Is this in a supplementary table?

Our response 9

We have modified Figure 3b by labeling the top 18 candidate genes. We refer the readers to Supplementary Dataset S1 for the complete list of candidate genes from the genetic screening. We have accordingly added a call out for Figure 3b to this sentence (Line235-236).

“These strains were retested in a square plate agar format, and 18 bacterial mutants exhibited consistent loss-of-*rbohD* mortality (Figure 3b).”

The labeling on the circle in Figure 3b is overlapping and not possible to read in places. This should be fixed.

Our response 10

We have addressed this. Please see Our response 7.

Reviewer #3 (Remarks to the Author):

The authors have substantially addressed my concerns. Just one minor point. Please fix the problems of overlapping text in Fig. 3b by labelling the gene names properly.

Our response 11

We appreciate Reviewer #3 comments to greatly improve our manuscript. We have modified Figure 3b by labeling the top 18 candidate genes. We refer the readers to Supplementary Dataset S1 for the complete list of candidate genes from the genetic screening.